# SAMD1 suppresses epithelial–mesenchymal transition pathways in pancreatic ductal adenocarcinoma

Clara Simon[1], Inka D. Brunke[1], Bastian Stielow[1], Ignasi Forné[2], Anna Mary Steitz[3], Merle Geller[1], Iris Rohner[1], Lisa Marie Weber[1], Sabrina Fischer[1], Lea Marie Jeude[1], Theresa Huber[1], Andrea Nist[4,5], Thorsten Stiewe[4,5], Magdalena Huber[6], Malte Buchholz[7], Robert Liefke [1,8]*

1 Institute of Molecular Biology and Tumor Research (IMT), Philipps University of Marburg, Marburg, Germany, 2 Protein Analysis Unit, Biomedical Center (BMC), Faculty of Medicine, Ludwig-Maximilians-University (LMU) Munich, Martinsried, Germany, 3 Translational Oncology Group, Center for Tumor Biology and Immunology (ZTI), Philipps University of Marburg, Marburg, Germany, 4 Genomics Core Facility, Institute of Molecular Oncology, Member of the German Center for Lung Research (DZL), Philipps University of Marburg, Marburg, Germany, 5 Institute for Lung Health (ILH), Justus Liebig University, Giessen, Germany, 6 Institute of Systems Immunology, Center for Tumor Biology and Immunology (ZTI), Philipps University of Marburg, Marburg, Germany, 7 Department of Gastroenterology, Endocrinology, Metabolism and Infection, Center for Tumor Biology and Immunology (ZTI), Philipps University of Marburg, Marburg, Germany, 8 Department of Hematology, Oncology, and Immunology, University Hospital Giessen and Marburg, Marburg, Germany

* robert.liefke@imt.uni-marburg.de

**Data Availability Statement:** ChIP-seq and RNA-seq data were uploaded to the Gene Expression

## Abstract

Pancreatic ductal adenocarcinoma (PDAC) poses a significant threat due to its tendency to evade early detection, frequent metastasis, and the subsequent challenges in devising effective treatments. Processes that govern epithelial—mesenchymal transition (EMT) in PDAC hold promise for advancing novel therapeutic strategies. SAMD1 (SAM domain-containing protein 1) is a CpG island-binding protein that plays a pivotal role in the repression of its target genes. Here, we revealed that SAMD1 acts as a repressor of genes associated with EMT. Upon deletion of SAMD1 in PDAC cells, we observed significantly increased migration rates. SAMD1 exerts its effects by binding to specific genomic targets, including *CDH2*, encoding N-cadherin, which emerged as a driver of enhanced migration upon SAMD1 knockout. Furthermore, we discovered the FBXO11-containing E3 ubiquitin ligase complex as an interactor and negative regulator of SAMD1, which inhibits SAMD1 chromatin-binding genome-wide. High *FBXO11* expression in PDAC is associated with poor prognosis and increased expression of EMT-related genes, underlining an antagonistic relationship between SAMD1 and FBXO11. In summary, our findings provide insights into the regulation of EMT-related genes in PDAC, shedding light on the intricate role of SAMD1 and its interplay with FBXO11 in this cancer type.

Omnibus (GEO) database, with the accession numbers GSE239415 and GSE239414, respectively. The mass spectrometry proteomics data have been deposited to the ProteomeXchange Consortium via the PRIDE [45] partner repository with the dataset identifier PXD044104. Public genome-wide data are available from the GEO repository with the accession numbers: GSM945261 (H3K4me3), GSM818826 (H3K27ac), GSM1010788 (RNA Polymerase II) and GSM1606403 (ATAC-Seq). All other relevant data are within the paper and its Supporting information files.

**Funding:** This project was supported by the German Research Foundation (Deutsche Forschungsgemeinschaft, DFG, 109546710, 416910386, 516068166), the Fritz Thyssen Foundation (Fritz Thyssen Stiftung, 10.20.1.005MN) and the German José Carreras Leukemia Foundation (José Carreras Leukämie-Stiftung, DJCLS 06 R/2022) to R.L. Open access funding is provided by the German Research Foundation (Deutsche Forschungsgemeinschaft, DFG, 416910386) and the Open Access Publishing Fund of Philipps-Universität Marburg. The funders played no role in study design, data collection and analysis, decision to publish, or preparation of the manuscript.

**Competing interests:** The authors have declared that no competing interests exist.

**Abbreviations:** AOD, amino oxidase domain; ChIP, chromatin immunoprecipitation; EMT, epithelial–mesenchymal transition; ES, embryonic stem; GSEA, gene set enrichment analysis; PCA, principal component analysis; PCL, polycomb-like protein; PDAC, pancreatic ductal adenocarcinoma; SAMD1, sterile alpha motif domain-containing protein 1; SD, standard deviation; TME, tumor microenvironment; WH, winged helix.

## Introduction

Pancreatic cancer is a highly lethal form of cancer, accounting for 2.8% of newly diagnosed cancer cases but contributing to 4.7% of cancer-related deaths [1]. Unlike many other cancer types, the incidence and mortality rates have steadily increased in recent years [2]. The most prevalent and severe subtype of pancreatic cancer is pancreatic ductal adenocarcinoma (PDAC), with a 5-year survival rate of only 9% in the United States [2].

Due to unspecific and late-occurring symptoms, PDAC is usually diagnosed in advanced stages [3]. Treatment options mainly include surgery and chemotherapy; however, most tumors are already deemed inoperable at diagnosis [4]. Epithelial—mesenchymal transition (EMT) is a crucial process in pancreatic cancer and is involved in early metastasis [5]. Genetically, PDAC is characterized by diverse mutations, with commonly affected genes including *TP53*, *CDKN2A*, *SMAD4*, and *KRAS* [6]. These genetic alterations contribute to the heterogeneity of tumors, which can vary substantially from patient to patient [6]. Recent advancements in PDAC research have focused on targeting the tumor microenvironment (TME). The TME in PDAC is characterized by its dense and desmoplastic nature, thereby influencing the druggability and chemoresistance of the tumor [7]. However, the identification of novel biomarkers for the early diagnosis of PDAC and the investigation of new druggable proteins are needed.

Sterile alpha motif domain-containing protein 1 (SAMD1) belongs to a novel class of CpG island-binding proteins alongside the histone acetyltransferases KAT6A and KAT6B [8–10]. These proteins share a winged helix (WH) domain that enables direct interaction with unmethylated CpG-rich DNA [8,10]. CpG islands (CGIs) are regulatory elements commonly found at promoter regions and play a critical regulatory role. Methylation of CGIs typically results in transcriptional silencing, while unmethylated CGIs are associated with active gene transcription [11]. CXXC domain-containing proteins and polycomb-like proteins (PCLs) have also been identified to interact specifically with unmethylated CGIs [12–15].

In mouse embryonic stem (ES) cells, SAMD1 was found to be present at thousands of unmethylated CGIs and to recruit the chromatin regulator L3MBTL3 and the histone demethylase KDM1A to its genomic targets, thereby acting as a transcriptional repressor [8]. Deletion of SAMD1 in mouse embryonic stem cells leads to the dysregulation of multiple cellular pathways, including neuronal, developmental, and immune response pathways [8]. The absence of SAMD1 during mouse embryogenesis primarily impairs brain development and angiogenesis and leads to embryonic lethality [16]. SAMD1 function is also linked to muscle adaptation after exercise [17] and autism spectrum disorders [18].

In multiple tumor types, *SAMD1* expression is up-regulated [9], and in liver cancer cells, knockout of SAMD1 has been shown to reduce proliferation and clonogenicity [19]. Moreover, liver cancer patients with high levels of SAMD1 exhibit a more unfavorable transcriptional network [19]. In the context of other cancer types, the role of SAMD1 remains largely unexplored.

Here, we show that in PDAC, SAMD1 acts as a repressor of EMT-related genes. After deleting SAMD1 in PDAC cell lines, we observed increased migration rates and up-regulation of cancer-associated pathways, including the EMT pathway. We identified *CDH2* as a key downstream target of SAMD1 that is important for the migration phenotype. Furthermore, we identified the E3 ubiquitin ligase F-box only protein 11 (FBXO11) as an interactor of SAMD1 in PDAC, which inhibits the chromatin association of SAMD1. Together, our study offers novel insights into the control of EMT-related genes in PDAC, revealing the intricate involvement of SAMD1 and its interplay with FBXO11 in this cancer type.

## Results

### SAMD1 regulates EMT pathways in PDAC

Investigation of public cancer gene expression data from TCGA showed that *SAMD1* is commonly up-regulated in cancer [20,21] (S1A Fig). In some cancer types, high *SAMD1* expression correlates with poor prognoses, such as in liver cancer (LIHC) and kidney renal clear cell carcinoma (KIRC), indicating a more oncogenic role (S1B Fig). In some other cancer types, such as cervical cancer (CESC) and thymoma (THYM), high *SAMD1* expression correlates with a better prognosis (S1C Fig), suggesting a more tumor-suppressive role. Interestingly, the distinct relationships to patient survival do not correlate with changes in *SAMD1* gene expression upon tumorigenesis, given that *SAMD1* has increased expression in most cancer tissues compared to normal tissues (S1A Fig). Thus, it is currently unknown why *SAMD1* has these potentially opposing roles in different cancer types.

A vital cancer type in which high *SAMD1* expression correlates with a better prognosis is PDAC (Fig 1A). The difference in survival becomes particularly evident at later time points, suggesting that a high SAMD1 expression level prevents an adverse cancer progression over time.

We hypothesized that gaining a deeper understanding of the potential tumor-suppressive role of SAMD1 in PDAC may allow us to employ this function to limit cancer growth. To acquire insights into the role of SAMD1 in PDAC, we investigated TCGA data using gene set enrichment analysis (GSEA) [23] and compared PDAC samples with high and low *SAMD1* expression. In the samples with high *SAMD1* expression, we found a strongly decreased expression of genes related to EMT (Fig 1B). Similar results were also obtained for thymoma and cervical cancer (S1D Fig), where high *SAMD1* expression also correlates with a better prognosis (S1B Fig).

In contrast, the opposite is the case in cancer types where high *SAMD1* correlates with a worse prognosis, such as kidney cancer. Here, the expression of EMT pathway genes positively correlates with *SAMD1* expression (S1B and S1D Fig). Notably, in all investigated cancer types, high *SAMD1* expression correlates with high expression of MYC target genes (S1E Fig), independent of whether high *SAMD1* expression is favorable or unfavorable, suggesting that this feature is not predictive. Together, these observations lead to the hypothesis that in some cancer types, such as PDAC, SAMD1 may be involved in repressing EMT pathways, thereby inhibiting metastasis, which in turn may contribute to a better outcome. In PDAC, EMT is particularly relevant because it strongly correlates with the occurrence of metastasis, which massively reduces the chance of survival [24].

To address the role of SAMD1 in the PDAC cells in more detail, we performed CRISPR/Cas9-mediated knockout approaches in the PDAC cell line PaTu8988t, validated via western and immunofluorescence (Fig 1C and 1D). We did not observe a change in the proliferation rate upon SAMD1 deletion (Fig 1E). Still, investigation of the cells under the microscope showed that SAMD1 knockout led to a more elongated cell shape and more pronounced protrusions (Fig 1F and 1G). This phenotype suggested an increased mobility of the knockout cells. By analyzing the migration rates of PaTu8988t cells by wound healing (Fig 1H and 1I), we confirmed the higher mobility of the SAMD1 KO cells. Furthermore, transwell migration through 8 μm pores demonstrated increased migration rates after SAMD1 deletion in PaTu8988t cells (S2A Fig), which could be visualized via crystal violet staining (S2B Fig). We confirmed these results via an unbiased approach by tracking PaTu8988t cells for 24 h using time-lapse analysis (S2C Fig and S1 and S2 Movies). Moreover, we tested the adhesion rates of control and SAMD1 KO cells to collagen, which were significantly higher in the SAMD1 KO cells (S2D Fig). In line with these findings, the SAMD1 KO cells also displayed higher invasion rates upon invasion through a collagen gel towards an FBS gradient (S2E and S2F Fig).

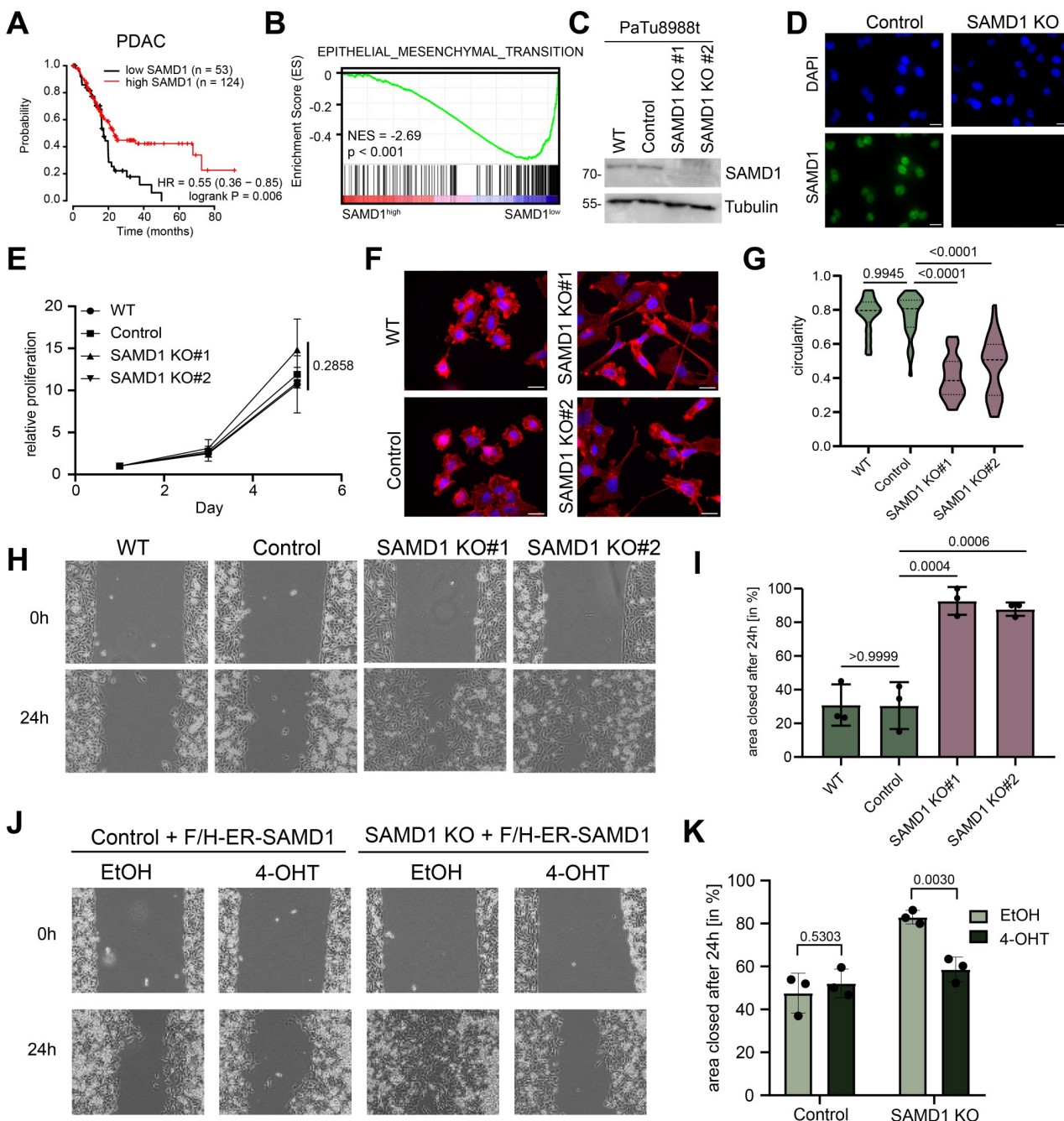

**Fig 1. SAMD1 inhibits EMT pathways and cell migration.** (A) Kaplan—Meier survival curves showing the correlation of *SAMD1* expression with patient survival. Graphs were visualized via KM plotter [22]. (B) GSEA for EMT using TCGA data analyzed for high and low *SAMD1* expression. (C) Western blot showing PaTu8988t wild-type cells, control cells, and 2 different SAMD1 knockout clones. (D) Immunofluorescence of PaTu8988t wild type, and SAMD1 knockout cells, Bar = 20 μm. (E) Proliferation assay of PaTu8988t wild-type, control cells, and 2 different SAMD1 knockout clones. Data represent the mean ± SD of 3 biological replicates. Significance was analyzed using one-way ANOVA. (F) Representative phalloidin staining of PaTu8988t wild-type cells, control cells, and 2 different SAMD1 knockout clones, Bar = 20 μm. (G) Cell shape of PaTu8988t wild-type cells, control cells, and 2 different SAMD1 knockout clones. Circularity was determined using ImageJ Fiji. Significance was analyzed using one-way ANOVA. (H) Representative picture of one wound healing assay of PaTu8988t wild-type cells, control cells, and 2 different SAMD1 knockout clones. (I) Quantification of the wound healing assay from (H). Data represent the mean ± SD of 3 biological replicates. Significance was analyzed using one-way ANOVA. (J) Representative picture of one wound healing assay of PaTu8988t control or SAMD1 KO cells with or without induction of SAMD1 rescue. (K) Quantification of the wound healing assay from (J). Data represent the mean ± SD of 3 biological replicates. Significance was analyzed using Student's *t* test. The data underlying this figure is available in S1 Data. EMT, epithelial—mesenchymal transition; GSEA, gene set enrichment analysis; SAMD1, sterile alpha motif domain-containing protein 1.

Enhanced cellular migration, but no change of proliferation, could also be found in BxPC3 PDAC cancer cells upon SAMD1 deletion (S3A–S3D Fig). These results suggest that the influence on the cellular properties by SAMD1 is not restricted to a single cell line but is a more general theme in PDAC, consistent with the anticorrelation of *SAMD1* expression and EMT pathways in PDAC patients (Fig 1B).

To investigate whether the observed phenotype depends on the nuclear function of SAMD1, we made use of an estrogen-receptor (ER)-SAMD1 fusion protein, whose nuclear localization can be induced by 4-hydroxy-tamoxifen (4-OHT) (S4A–S4C Fig). Using this approach, we demonstrated that the migration phenotype in PaTu8988t SAMD1 KO cells can be rescued upon translocation of SAMD1 into the nucleus (Fig 1J and 1K), suggesting that the observed phenotype is linked to the chromatin regulatory role of SAMD1. Importantly, the estrogen receptor alone had no impact on the migration rates of the cells (S4D and S4E Fig).

## SAMD1 directly represses CDH2, a key regulator of EMT

To address which SAMD1 target genes participate in this phenotype, we performed gene expression analysis via RNA-seq upon SAMD1 knockout and analyzed the genomic distribution of SAMD1 via ChIP-seq in PaTu8988t cells. Principal component analysis (PCA) of the RNA-seq data demonstrated that the knockout led to a substantial shift in the transcriptional landscape (S5A Fig). The knockout of SAMD1 led to significant deregulation of 854 genes, with 642 up-regulated and 212 down-regulated genes (cut-off: log2-fold-change $> 0.5$; *p*-value $< 0.01$) (Figs 2A and S5B). GSEA analysis of the RNA-seq data demonstrated that the deletion of SAMD1 leads to the dysregulation of multiple cancer-related pathways (Fig 2B). Specifically, we observed an up-regulation of signaling pathways, including Hedgehog, KRAS, and WNT, and a down-regulation of MYC and E2F target genes. Additionally, many transcription factors, including HOXB cluster genes, become dysregulated upon SAMD1 deletion (S5B Fig). Importantly, EMT pathway genes showed a trend for up-regulation in the knockout cells (Fig 2C). Also, genes related to mesenchymal proliferation and mesenchyme morphogenesis were significantly up-regulated (Figs 2C, S5C and S5D). Within these gene sets many key EMT-related genes such as *CDH2* [25], *BMP2, BMP4, BMP7* [26], Netrin (*NTN1*) [27], *SOX9* [28], *WNT5A* [29], *WNT11, VCAN,* and *ZEB1* [30] showed significant up-regulation in the SAMD1 KO PaTu8998 cells (S5E Fig). Similar results were also seen in BxPC3 and CFPAC PDAC cells lines (S5F Fig). In addition, we found the down-regulation of *TJP1* (Tight junction protein 1) in PaTu8988t cells, suggesting a loss of cell—cell contacts in the SAMD1 KO cells (S5G Fig). These findings are consistent with the observed phenotype and in line with our initial hypothesis that SAMD1 may be involved in regulating mesenchymal pathways in pancreatic cancer cells.

Notably, there is hardly any correlation between the gene expression changes upon SAMD1 deletion in distinct cell types, as demonstrated by the comparison of the gene expression changes between PaTu8988t cells and mouse ES cells [8] or HepG2 cells [19], respectively (S5H Fig). Only the *L3mbtl3/L3MBTL3* gene is consistently dysregulated upon SAMD1 deletion (S5H Fig). In line with these distinct consequences on gene expression, the EMT pathways become only up-regulated in PaTu8988t cells, but not in mouse ES and HepG2 cells (S5I Fig). These findings underline the strong context dependency of SAMD1 function on gene regulation and that the influence of SAMD1 on EMT pathways may only occur in some cell types, such as pancreatic cancer cells.

Genome-wide analysis of SAMD1 chromatin binding showed that SAMD1 mainly binds to CpG island-containing gene promoters (Fig 2D), which are in an active state, characterized by the enrichment of active histone marks, accessible chromatin and RNA-polymerase II (S6A

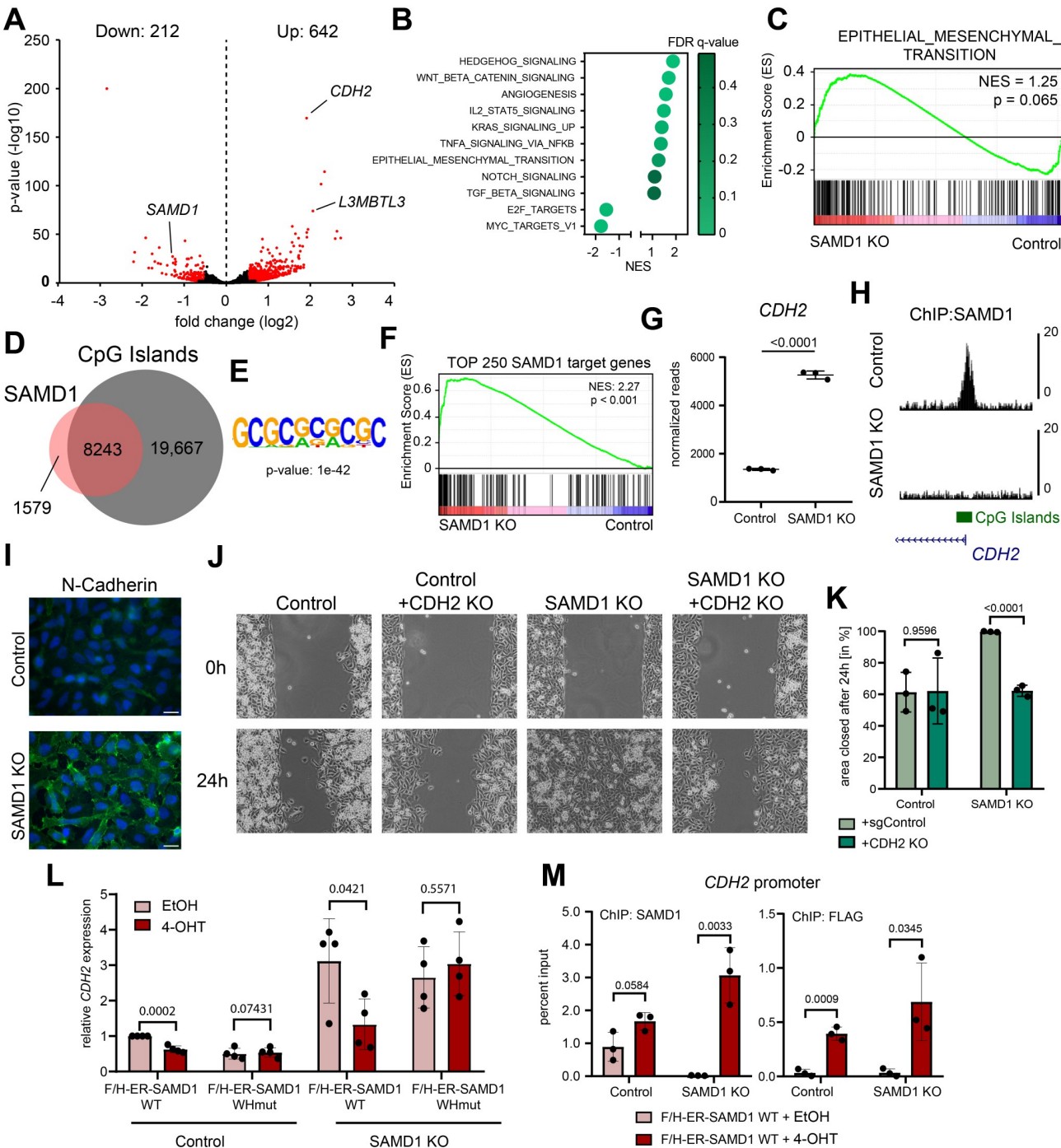

**Fig 2. SAMD1 directly regulates EMT pathway genes in PaTu8988t cells.** (A) Volcano plot of RNA-seq data comparing the results from 3 replicates of PaTu8988t control cells with 3 clonally independent SAMD1 KO cells. (B) GSEA for several pathways comparing the results from 3 replicates of PaTu8988t control cells with 3 clonally independent SAMD1 KO cells. (C) GSEA of EMT from (B). (D) Venn diagram showing the overlap of SAMD1 peaks with all CpG islands in PaTu8988t cells. (E) Enriched motif at SAMD1-bound CGIs. (F) GSEA of the top 250 SAMD1 targets in PaTu8988t cells, comparing the results from 3 replicates of control cells with 3 clonally independent SAMD1 KO cells. (G) RNA-seq results for *CDH2* expression (RPKM) comparing the results from 3 replicates of PaTu8988t control cells with 3 clonally independent SAMD1 KO cells. (H) Snapshot of the USCS browser showing a SAMD1 peak at the *CDH2* promoter in PaTu8988t control and SAMD1 KO cells. (I) Immunofluorescence of N-cadherin in PaTu8988t control and SAMD1 KO cells, Bar = 20 µm. (J) Representative picture of one wound healing assay of PaTu8988t control, CDH2 KO, SAMD1 KO, and CDH2/SAMD1 double KO cells. (K) Quantification of the wound healing assay from (J). Data represent the mean ± SD of 3 biological replicates. Significance was analyzed using Student's *t* test. (L) RT-qPCR showing *CDH2* expression with or without induction of SAMD1 rescue in PaTu8988t control and SAMD1 KO cells. WHmut = R45A/K46A mutation of SAMD1 [8]. Data represent the mean ± SD of 4 biological replicates. Significance was

analyzed using Student's *t* test. (M) SAMD1 ChIP-qPCR at the *CDH2* promoter with or without induction of SAMD1 rescue in PaTu8988t control and SAMD1 KO cells. Data represent the mean ± SD of 3 biological replicates. Significance was analyzed using Student's *t* test. The data underlying this figure is available in S1 Data. EMT, epithelial—mesenchymal transition; GSEA, gene set enrichment analysis; SAMD1, sterile alpha motif domain-containing protein 1.

Fig). SAMD1-bound CpG islands have a GCGC motif enriched, consistent with SAMD1's pre-ferred binding motif [8] (Fig 2E). SAMD1 target genes are predominantly linked to chromatin and transcriptional regulation (S6B Fig) and a comparison of gene expression changes with SAMD1 promoter binding revealed that SAMD1 binds particularly strongly at genes that become up-regulated (S6C and S6D Fig). Consequently, the top SAMD1 target genes were sig-nificantly up-regulated upon SAMD1 deletion, as assessed by GSEA (Fig 2F). These results are largely consistent with our previous findings from mouse ES [8] and HepG2 cells [19] and sup-port that SAMD1 acts as a repressor at active CGIs. Consequently, we hypothesized that the increased migratory ability of the knockout cells may be established by the derepression of one or several SAMD1 target genes.

Within the EMT-related genes, *CDH2* was one of the top up-regulated factors in the SAMD1 KO cells (Fig 2A and 2G). Its promoter also showed high SAMD1 enrichment (Fig 2H), suggesting a direct regulation by SAMD1. *CDH2* encodes for N-cadherin, a crucial regu-lator of cell adhesion and consequently for EMT in PDAC [25,31]. We confirmed the up-regu-lation of N-cadherin via immunofluorescence (Fig 2I). Therefore, *CDH2* could be a key downstream target of SAMD1 in PaTu8988t cells to influence cellular migration. Indeed, the cells possessing a double knockout of CDH2 and SAMD1 had similar characteristics to the wild-type cells regarding their migratory ability (Fig 2J and 2K) and cellular shape (S7A and S7B Fig). Additionally, short-term inhibition of N-cadherin—mediated cell adhesion by ADH-1 (Exherin) [32] reduced the elongated phenotype of SAMD1 KO cells, making them more similar to wild-type cells (S7C and S7D Fig). These findings support that *CDH2* is a critical downstream factor of SAMD1 that regulates the migration properties of the PaTu8988t cells.

To assess whether SAMD1 directly regulates *CDH2*, we used the ER-SAMD1 fusion protein described above. In SAMD1 KO cells expressing this fusion protein, the expression of *CDH2* was rescued upon 4-OHT treatment (Fig 2L). This rescue does not work with a winged helix domain mutant of SAMD1, indicating that the chromatin binding of SAMD1 is essential for the repression of *CDH2*. Furthermore, ChIP-qPCR confirmed that SAMD1 chromatin binding to the *CDH2* promoter can be rescued with the ER-SAMD1 fusion protein (Fig 2M). Similar results were also obtained for the *L3MBTL3* gene, a known SAMD1 target gene that becomes consistently up-regulated upon SAMD1 deletion in several distinct cell types [8,19] (S5G, S8A and S8B Figs). The chromatin binding and regulatory effect of SAMD1 on *CDH2* and *L3MBTL3* was also confirmed in BxPC3 cells (S8C and S8D Fig).

Together, these results suggest that SAMD1 is directly involved in repressing *CDH2* in PDAC cells, an essential regulator of EMT [25].

## The repressive activity of SAMD1 likely involves KDM1A

The molecular details of the repressive function of SAMD1 are currently not fully understood. Previously, we showed that SAMD1 interacts with the KDM1A complex, which demethylates the active H3K4me2 histone mark, and the SAM- and MBT-domain proteins L3MBTL3 and SFMBT1 [8]. In the context of PaTu8988t cells, we confirmed that in the absence of SAMD1, the levels of KDM1A and L3MBTL3 are reduced at the *CDH2* gene, which can be rescued upon induction of the ER-SAMD1 with 4-OHT (Fig 3A). Consistently, via ChIP-qPCR, we observed an increase in H3K4me3 and a slightly increased signal for H3K4me2 at the

*L3MBTL3* and *CDH2* gene promoters upon SAMD1 deletion (Fig 3B). We also validated the interaction between SAMD1 and KDM1A in PaTu8988t cells via endogenous co-immunoprecipitation (Fig 3C). These results support that in PaTu8988t cells, KDM1A is likely involved in the repressive function of SAMD1.

To gain further insight into the interplay of SAMD1 with the KDM1A complex (Fig 3D), we performed mapping experiments that went beyond our previous experiments [8]. First, we confirmed that SAMD1 preferentially interacts with KDM1A via its winged helix domain and that deleting the SAM domain, which is essential for the interaction with L3MBTL3 [8], increases the interaction with KDM1A (Fig 3E). This phenomenon is also observable with other members of the KDM1A complex (S9A Fig). Reverse mapping experiments suggested that SAMD1 preferentially interacts with the N-terminal part of the catalytic amino oxidase domain (AOD) of KDM1A (Fig 3F). This result is further supported by the observation that the KDM1A inhibitor ORY-1001 (Iadademstat), which covalently binds to the FAD cofactor within KDM1A [33], interferes with the interaction of KDM1A with SAMD1 (S9B Fig). This effect was not observed for other KDM1A complex members, such as RCOR1 and PHF21A (S9C and S9D Fig). This finding raises the possibility that SAMD1 may bind near the catalytic cleft of KDM1A, potentially causing an increased sensitivity of the interaction to the inhibitor treatment.

Based on these results, we speculated that SAMD1 could potentially influence the demethylase activity of KDM1A, similar to other factors associated with the KDM1A complex [34]. To address this question, we immunoprecipitated KDM1A in either wild-type or SAMD1 KO HEK293 cells and used the obtained precipitate for demethylase assays, adding calf histones as a substrate. We found that the absence of SAMD1 reduced the ability of KDM1A to remove H3K4me2 efficiently (Fig 3G and 3H). This result suggests that SAMD1 modulates the function of KDM1A, possibly not just by influencing its recruitment to chromatin but also by influencing the catalytic activity of KDM1A. Both processes together may contribute to the repressive role of SAMD1. However, SAMD1's association with KDM1A may not only influence the function of KDM1A itself but may also affect KDM1A-associated factors, such as RCOR1, which in turn may be important for the chromatin association of the KDM1A complex. Thus, the molecular details of how exactly SAMD1 affects the enzymatic activity and chromatin binding of KDM1A require further research. In addition, we cannot exclude the possibility that other mechanisms, such as the recruitment of L3MBTL3 [8], are also crucial for the repressive function of SAMD1.

## SAMD1 interacts with the FBXO11 E3 ubiquitin ligase complex

Upon investigating the cellular localization of SAMD1 in PaTu8988t cells and further PDAC cell lines, we observed that compared to other human cell lines, SAMD1 is less present in the chromatin fraction in PDAC cells (Fig 4A). This finding raises the possibility that a certain molecular mechanism regulates the chromatin binding of SAMD1. In the context of pancreatic cancer cells, such a mechanism may be essential to overcome the tumor-suppressive function of SAMD1. To date, no process has been described that regulates the chromatin association of SAMD1.

To address whether SAMD1 may interact with additional proteins that could be involved in such a regulatory process, we performed unbiased IP-MS experiments in PaTu8988t cells. For this, we used cells expressing the ER-SAMD1 fusion protein (S4A–S4C Fig). After inducing the nuclear localization of the protein via 4-OHT, we collected the cells and immunoprecipitated the SAMD1 protein. The co-bound proteins were analyzed by LC-MS (Fig 4B). This experiment confirmed that SAMD1 interacts with L3MBTL3 and the KDM1A histone

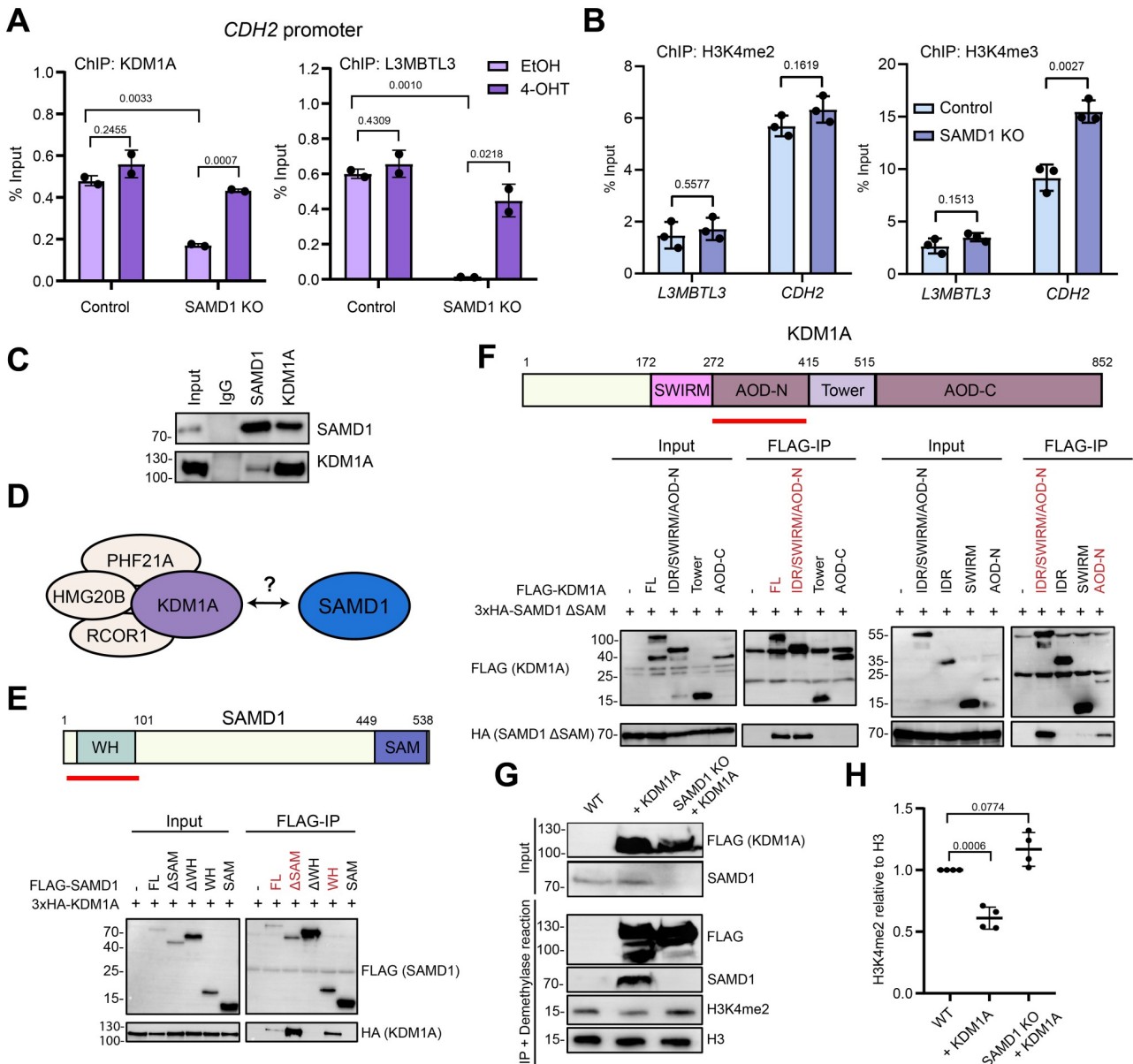

**Fig 3. SAMD1 is required for full activity of KDM1A.** (A) ChIP-qPCR at the *CDH2* promoter with or without induction of SAMD1 rescue in PaTu8988t control and SAMD1 KO cells using KDM1A and L3MBTL3 antibodies. Data represent the mean ± SD of 2 biological replicates. Significance was analyzed using Student's *t* test. (B) ChIP-qPCR of *CDH2* and *L3MBTL3* promoter in PaTu8988t control and SAMD1 KO cells using H3K4me2 and H3K4me3 antibodies. Data represent the mean ± SD of 3 biological replicates. Significance was analyzed using Student's *t* test. (C) Western blot of an endogenous Co-IP between SAMD1 and KDM1A in PaTu8988t cells. (D) Model of the interaction between SAMD1 and the KDM1A complex. (E) Structure of SAMD1; co-immunoprecipitation in HEK293 cells showing the interaction between different SAMD1 deletion mutants and KDM1A. Regions identified to interact with KDM1A are marked red. (F) Structure of KDM1A; co-immunoprecipitation in HEK293 cells showing the interaction between different KDM1A deletion mutants and SAMD1. Regions identified to interact with SAMD1 are marked red. (G) Representative western blot of KDM1A IP in HEK293 cells, followed by histone demethylase assay. (H) Quantification of 4 biological replicates of (G). Significance was analyzed using one-way ANOVA. The data underlying this figure is available in S1 Data. ChIP, chromatin immunoprecipitation; SAMD1, sterile alpha motif domain-containing protein 1.

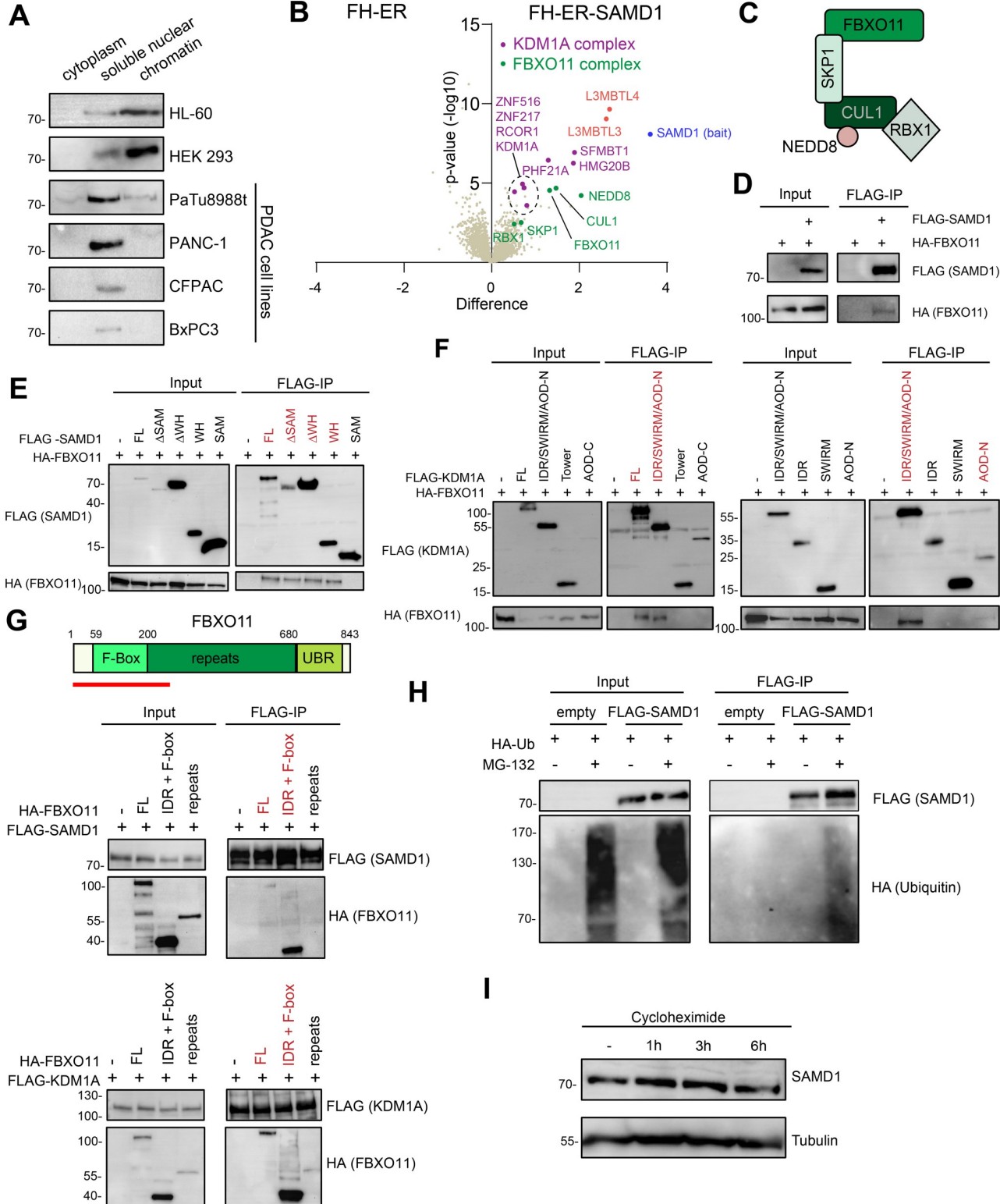

**Fig 4. SAMD1 interacts with FBXO11 in PaTu8988t cells and is ubiquitinated.** (A) Fractionation of different cell lines followed by SAMD1 western blotting. (B) Volcano plot of proteins identified by mass spectrometry after IP of FH-ER and FH-ER-SAMD1. (C) Model of the FBXO11 E3-ubiquitin ligase complex. (D) Co-immunoprecipitation in HEK293 cells showing the interaction between SAMD1 and FBXO11. (E) Co-immunoprecipitation in HEK293 cells showing the interaction between different SAMD1 deletion mutants and FBXO11. (F) Co-immunoprecipitation in HEK293 cells showing the interaction between different KDM1A deletion mutants and FBXO11. (G) Co-immunoprecipitation in HEK293 cells showing the interaction

between different FBXO11 deletion mutants with SAMD1 and KDM1A. (H) Ubiquitination assay after empty vector or SAMD1 transfection in HEK293 cells. (I) Cycloheximide chase analysis of SAMD1 in PaTu8988t cells. The data underlying this figure is available in S1 Data. SAMD1, sterile alpha motif domain-containing protein 1.

demethylase complex. We also identified L3MBTL4, consistent with our finding that the SAM domain of L3MBTL4 can interact with the SAM domain of SAMD1, similar to L3MBTL3 [8]. In addition to these expected interactions, we identified members of the FBXO11 complex as putative novel interaction partners of SAMD1. The FBXO11 complex consists of FBXO11 itself, RBX1, Cullin 1, and SKP1, all of which are enriched in the SAMD1 IP (Fig 4B and 4C). Additionally, we found enrichment of NEDD8 (Fig 4B), which is typically covalently associated with Cullin 1 (Fig 4C) and is required for the F-box protein-associated E3 ubiquitin ligase complexes to be active [35].

Via co-immunoprecipitation experiments in HEK293 cells, we validated that SAMD1 can interact with FBXO11 (Fig 4D). Additional mapping experiments suggested that several regions of SAMD1 are relevant for this interaction (Fig 4E). Only the SAM domain appears dispensable for the interaction with FBXO11 (Fig 4E). Interestingly, we found that FBXO11 can also be co-immunoprecipitated with KDM1A (Fig 4F). This interaction is mostly facilitated by the N-terminal part of KDM1A, which includes the N-terminal unstructured region and the N-terminal part of the AOD domain. A similar region is involved in the interaction with SAMD1 (Figs 3F and 4F), suggesting that KDM1A interacts with SAMD1 and FBXO11 simultaneously. Consistently, the interaction between KDM1A and FBXO11 is also sensitive to the KDM1A inhibitor ORY-1001 (S9E Fig), similar to the SAMD1/KDM1A interaction (S9B Fig). Reverse mapping experiments suggest that particularly the N-terminal part, including the F-box domain, of FBXO11 is relevant for the interaction with both SAMD1 and KDM1A (Fig 4G). In contrast, the UBR domain, which is unique for FBXO11 [36], is dispensable for these interactions (S9F and S9G Fig).

FBXO11 is an E3 ubiquitin ligase that regulates the ubiquitination of various target proteins, including BCL6, CDT2, and BAHD1 [37–39]. We hypothesized that FBXO11 may facilitate the ubiquitination of SAMD1. To investigate whether SAMD1 is ubiquitinated, we co-immunoprecipitated FLAG-tagged SAMD1 in HEK293 cells expressing HA-tagged ubiquitin. We observed that in the presence of the proteasome inhibitor MG-132, the immunoprecipitated FLAG-SAMD1 but not the control precipitate showed an HA-signal in western blotting experiments (Fig 4H), demonstrating that SAMD1 becomes ubiquitinated when the proteasome is inhibited. Surprisingly, however, cycloheximide chase experiments suggested that SAMD1 is highly stable, with no obvious turn-over within 6 h (Fig 4I).

## FBXO11 influences SAMD1 chromatin binding genome-wide

To assess which region of SAMD1 is mostly ubiquitinated, we used SAMD1 deletion mutants. We found that deleting the WH domain decreased the ubiquitination level, while deletion of the SAM domain, or using the WH domain alone led to an increased ubiquitination level (Fig 5A). This result suggests that the DNA binding WH domain is the primary ubiquitination site of SAMD1.

To investigate the role of FBXO11 in the ubiquitination of SAMD1, we established HEK293 and PaTu8988t cells with FBXO11 knockout (Fig 5B). Consistent with the idea that FBXO11 ubiquitinates SAMD1, we found a reduced level of ubiquitination of the SAMD1 WH domain in the FBXO11 knockout HEK293 cells (Fig 5C and 5D).

Notably, after FBXO11 knockout in PaTu8988t cells, we did not observe an altered protein level of SAMD1 (Fig 5E), suggesting that ubiquitination of SAMD1 by FBXO11 has no relevant

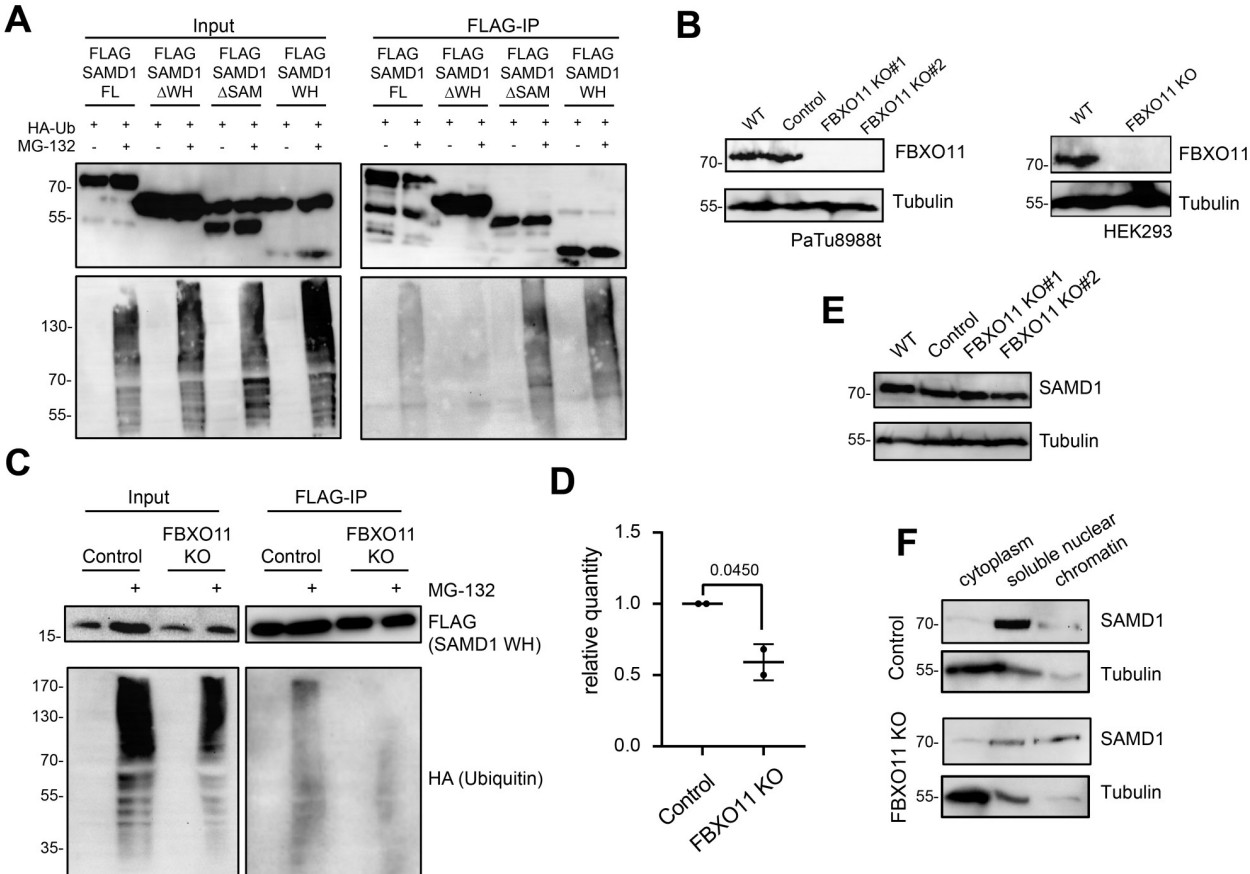

**Fig 5. FBXO11 affects SAMD1 ubiquitination and chromatin association.** (A) Ubiquitination assay in HEK293 cells using various SAMD1 deletion constructs. (B) Western blot showing PaTu8988t wild-type cells, control cells, and 2 different FBXO11 knockout clones; western blot showing HEK293 cells with FBXO11 KO. (C) Ubiquitination assay of SAMD1 WH domain in HEK293 control and FBXO11 KO cells. (D) Quantification of (C) using 2 biological replicates. (E) Western blot showing SAMD1 expression in PaTu8988t wild-type cells, control cells, and 2 different FBXO11 knockout clones. (F) Fractionation of PaTu8988t control and FBXO11 KO cells, followed by SAMD1 western blotting. The data underlying this figure is available in S1 Data. SAMD1, sterile alpha motif domain-containing protein 1; WH, winged helix.

influence on the turn-over of SAMD1. However, fractionation experiments showed that the chromatin association of SAMD1 was substantially augmented in the FBXO11 KO cells (Fig 5F), suggesting that the FBXO11 is involved in modulating the chromatin association of SAMD1.

To address the role of FBXO11 in SAMD1 chromatin binding in further detail, we performed ChIP-qPCR using our SAMD1 antibody. Consistent with the fractionation experiment (Fig 5F), we observed an enhanced chromatin binding of SAMD1 at the *CDH2* promoter in the FBXO11 KO PaTu8988t cells (Fig 6A). Although the result is not significant, we observed the increase in 3 independent experiments. To investigate this effect at the genome-wide level, we performed ChIP-seq of SAMD1 in wild-type and FBXO11 KO cells in 2 biological replicates. We found an increased SAMD1 binding FBXO11 knockout cells in both experiments (Figs 6B and S10A), supporting that FBXO11 is involved in inhibiting the chromatin binding of SAMD1 at a global level.

The impact of FBXO11 deletion was particularly evident at locations where SAMD1 chromatin binding was low under wild-type conditions (Fig 6C and 6D, groups 1 and 2, and S10B Fig). In contrast, in places where SAMD1 was already strongly present in the wild-type cells,

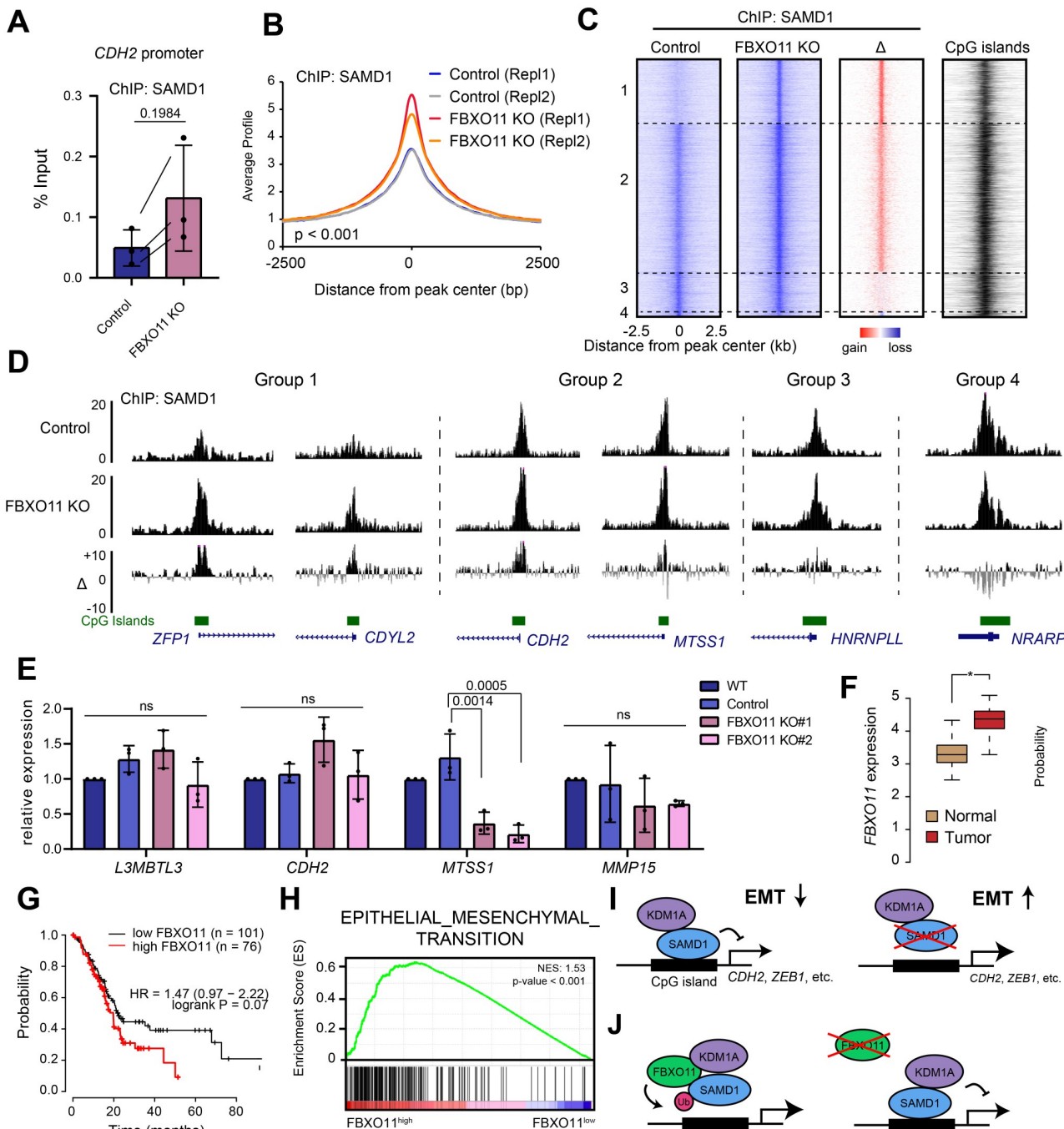

**Fig 6. FBXO11 counteracts SAMD1.** (A) ChIP-qPCR at the *CDH2* promoter in PaTu8988t control and FBXO11 KO cells, using a SAMD1 antibody. Data represent the mean ± SD of 3 biological replicates. (B) Profiles of SAMD1 at SAMD1 peaks, in wild-type and FBXO11 KO cells. Significance was evaluated via a Kolmogorov—Smirnov test. See also S10A Fig. (C) Heatmaps of SAMD1 ChIP-Seq results (first replicate) at all SAMD1 peaks in PaTu8988t control and FBXO11 KO cells. Peaks were grouped according to the gain or loss after FBXO11 KO (Group 1, *n* = 3,028; Group 2, *n* = 6,895; Group 3, *n* = 1,883; Group 4, *n* = 229); Δ indicates the difference between FBXO11 KO and control cells. SAMD1 peaks overlap with CpG islands. See also S10B Fig. (D) Snapshots of ChIP-Seq results (first replicate) in the USCS browser showing examples of the 4 groups presented in (C). (E) RT-qPCR analysis of SAMD1 target genes with enhanced SAMD1 chromatin binding upon FBXO11 KO. Data represent the mean ± SD of 3 biological replicates. Significance was analyzed using one-way ANOVA. ns = not significant. (F) Expression of FBXO11 in PDAC versus normal tissues. Data from TCGA [20] and visualized via GePIA [21]. (G) Kaplan—Meier survival curve showing the correlation of *FBXO11* expression with patient survival. Graph was visualized via KM plotter [22]. (H) GSEA for EMT using TCGA data analyzed for high and low FBXO11 expression. (I) Model of the role of SAMD1 in PDAC. SAMD1 represses EMT-related genes, thereby suppressing migration. In SAMD1 KO cells, the EMT genes become up-regulated leading to an enhanced EMT-related phenotype. (J) FBXO11 is counteracting SAMD1 chromatin association, possibly via ubiquitination processes. Absence of

FBXO11 leads to enhanced chromatin binding of SAMD1. The data underlying this figure is available in S1 Data. ChIP, chromatin immunoprecipitation; EMT, epithelial—mesenchymal transition; GSEA, gene set enrichment analysis; PDAC, pancreatic ductal adenocarcinoma; SAMD1, sterile alpha motif domain-containing protein 1.

the FBXO11 knockout had only minor effects (Fig 6C and 6D, group 3, and S10B Fig). Only at a very small fraction, with very high SAMD1 levels, SAMD1 occupancy becomes weaker (Fig 6C and 6D, group 4, and S10B Fig). Closer inspection of the distinct groups showed that the locations more susceptible to FBXO11 deletion have smaller CpG islands (S10C Fig) and are transcriptionally more active, signified by higher H3K4me3 and RNA Polymerase II levels (S10D and S10E Fig). This observation raises the possibility that FBXO11 regulates SAMD1 chromatin binding more strongly at locations with fewer CpG binding motifs and higher transcriptional activity. On the other hand, no substantial differences regarding the gene expression levels of the associated genes could be recognized (S10F Fig).

To investigate whether the enhanced binding of SAMD1 contributes to changes in gene expression, we analyzed several SAMD1 target genes via RT-qPCR. We found a significant reduction in *MTSS1* gene expression but no changes in other investigated genes (Fig 6E). This observation suggests that the increased chromatin binding of SAMD1 upon FBXO11 deletion influences the gene expression of some specific genes, but not of others. The approximately 50% elevation of the chromatin binding level of SAMD1 is possibly not sufficient to substantially influence the gene expression of most genes. Consistently, we did not observe an obvious migration phenotype of the FBXO11 KO PaTu8988t cells (S10G and S10H Fig).

To address whether overexpressing FBXO11 has the opposite effect on SAMD1 chromatin binding, we first ectopically express FBXO11 in PaTu8988t cells. However, we were unsuccessful in gaining FBXO11 overexpressing PaTu8988t cells, since this leads to cell death. In an alternative model, we performed the experiment in HEK293 cells, where we could successfully express FBXO11. Using these cells, we found that high FBXO11 leads to approximately 50% decreased SAMD1 chromatin levels at all investigated SAMD1 target genes, as assessed by ChIP-qPCR (S11A Fig). Notably, using an antibody against FBXO11, we found that FBXO11 also shows some degree of chromatin binding (S11A Fig), supporting the idea that FBXO11 affects SAMD1 directly at the chromatin. Investigation of the associated genes regarding gene expression showed only minimal impact on gene expression (S11B Fig), suggesting again that the observed change in chromatin binding of SAMD1 is not sufficient to strongly impair the gene expression of individual genes.

Our biochemical and genome-wide data support the hypothesis that FBXO11 interacts with and ubiquitinates SAMD1, which may impair its chromatin association and thereby could inhibit the gene repressive function of SAMD1. This mechanism may be important during cancer progression to reduce the tumor-suppressive role of SAMD1. This idea is supported by the fact that *FBXO11* is commonly up-regulated in PDAC (Fig 6F), and its high expression correlates with worse patient prognosis (Fig 6G). Furthermore, a high *FBXO11* expression level was linked to increased expression of EMT pathway genes (Fig 6H), which is opposite to what we observed before for *SAMD1* (Fig 1B). These opposing roles of SAMD1 and FBXO11 can also be seen for other cancer types, such as thymoma and cervical cancer (S12A and S12B Fig). Together, these data support the hypothesis that in PDAC, and perhaps other cancer types, the tumor-suppressive function of SAMD1 is counteracted by FBXO11.

## Discussion

PDAC is a highly lethal form of cancer [4]. Local invasion and metastasis, driven by uncontrolled EMT, are the main reasons for the aggressive nature of PDAC [24]. Unfortunately,

current strategies for inhibiting EMT in PDAC are not sufficiently effective. In this work, we identified the chromatin regulator SAMD1 as an important suppressor of EMT-related pathways in PDAC.

Analysis of patient data revealed that *SAMD1* is frequently dysregulated in cancer, and its expression often correlates with favorable or unfavorable prognoses. In the context of PDAC samples, *SAMD1* expression is commonly up-regulated, (S1A Fig), but Kaplan—Meier survival curves demonstrated that high SAMD1 expression is associated with a better outcome (Fig 1A), suggesting a tumor-suppressive role for SAMD1. Biological assays utilizing SAMD1 KO cells were conducted to investigate this further, revealing increased migration rates and up-regulation of EMT-related pathways upon SAMD1 loss (Figs 1H, 1I, and 6I). These findings suggest that lower SAMD1 expression levels may contribute to increased metastatic potential, thereby impacting patient survival rates, especially in later stages of the disease (Fig 1A). Consequently, the level of SAMD1 expression in pancreatic cancer samples at the time of diagnosis could potentially serve as a predictive marker for disease progression.

Via RNA-Seq, we identified *CDH2* as one of the top downstream targets of SAMD1 in PaTu8988t cells (Figs 2 and 3A). We show that *CDH2*, the gene encoding N-cadherin, serves as the main driver of enhanced migration after SAMD1 KO (Fig 2J and 2K). During EMT progression, cadherins play a central role. The down-regulation of epithelial cadherin (E-cadherin), which is critical for adherens junction formation, is accompanied by an elevation in neural cadherin (N-cadherin) expression. This shift contributes to heightened cell mobility and a more mesenchymal phenotype [40]. While N-cadherin is only detectable in nearly 50% of all PDAC patient samples, its presence in metastatic lesions significantly correlates with augmented neural invasion and a higher histological grade [25]. Remarkably, tumors exhibiting high N-cadherin levels also show elevated *TGFB* expression, which resonates with our observations indicating the up-regulation of TGFB-signaling consequent to SAMD1 knockout (Fig 2B). The repression of CDH2 may therefore be a key element of SAMD1's tumor-suppressive role in PDAC. Besides *CDH2*, many other EMT-related genes, including *BMP2* [41], Netrin-1 (*NTN1*) [27], *WNT5A* [29], and *ZEB1* [30], are targeted and repressed by SAMD1 (Fig 6I).

During this work, we noticed a decreased chromatin binding of SAMD1 in PDAC cell lines compared to other tumor cell lines (Fig 4A). This finding led us to speculate that SAMD1 chromatin association is actively inhibited. Via IP-MS experiments, we revealed the FBXO11-containing E3-ubiquitin ligase complex as a novel interactor of SAMD1 (Fig 4B). FBXO11 plays a versatile role in cancer, acting both as an oncogene and as a tumor suppressor. It targets oncogenic proteins, such as BCL-6 or the Snail family of transcription factors, for degradation, thereby exhibiting a tumor-suppressive function in various cancer types, such as diffuse large B-cell lymphomas and lung cancer [37,42]. In lung cancer cell lines, the FBXO11-containing complex was found to neddylate p53, thereby inhibiting its transcriptional activity [43]. In contrast, in pancreatic ductal adenocarcinoma, silencing FBXO11 suppresses tumor development [44], which may involve the ubiquitination of p53, supporting an oncogenic role in this cancer type. Our work suggests that FBXO11 inhibits SAMD1 chromatin association, which represents a novel regulatory mechanism of FBXO11 in cancer (Fig 6J). A possible mechanism of how FBXO11 affects SAMD1 chromatin binding could be that ubiquitination of SAMD1's WH domain inhibits the DNA binding function of the domain. However, it is also possible that the interaction of FBXO11 with SAMD1 directly prevents chromatin binding, independent from ubiquitination. We currently can also not exclude more indirect mechanisms. Notably, although FBXO11 influences the chromatin binding of SAMD1, the consequences on gene expression appear rather subtle. Nonetheless, high FBXO11 expression is associated with a worse prognosis in PDAC (Fig 6D) [44], opposite to SAMD1 (Fig 1A). This result supports the notion that FBXO11 has an oncogenic role in PDAC, which may be at least in part be

linked to its chromatin-inhibiting role for SAMD1. More research will be necessary to clarify the detailed processes of how the FBXO11 E3 ubiquitin ligase complex influences SAMD1 function in the cells, and how this role contributes to the overall function of FBXO11 in cancer cells.

In our IP-MS experiments, we successfully identified another key interactor of SAMD1—the KDM1A histone demethylase complex (Fig 4B). KDM1A, also referred to as LSD1, has previously been characterized as an interactor of SAMD1 [8]. In mouse embryonic stem cells, the deletion of SAMD1 results in a reduction of KDM1A binding on specific promoters [8], implying that SAMD1 plays a role in the recruitment of KDM1A to chromatin. We confirmed that KDM1A is also an interactor of SAMD1 in PDAC (Fig 3C) and that deletion of SAMD1 leads to reduced recruitment of KDM1A to chromatin (Fig 3A) and altered H3K4 methylation levels (Fig 3B). Besides the involvement of SAMD1 in the recruitment of KDM1A, our work supports the idea that the presence of SAMD1 in the KDM1A complex influences the catalytic activity and the chromatin association of KDM1A (Figs 3A, 3G, 3H, and 6I). One could speculate that the association of SAMD1 with KDM1A affects the conformation of the KDM1A complex, which in turn may allow a more efficient demethylation reaction. Another possibility is that SAMD1, when bound to the KDM1A complex, enhances the association of the KDM1A complex with its nucleosomal substrate, which increases the efficiency of demethylation. It is also possible that the association of the FBXO11 complex with KDM1A contributes to the altered demethylase activity of KDM1A. It will be of interest to decipher the potentially sophisticated interplay of KDM1A, SAMD1, and FBXO11.

Based on our RNA-seq experiment performed here (Fig 2A) and previously [8,19], most SAMD1 target genes are only subtly influenced by SAMD1 deletion, leading to less than 10-fold up-regulation. Small changes of SAMD1 chromatin binding, such as due to regulation by FBXO11, may therefore only have small effects on individual genes. However, given that SAMD1 targets thousands of genes (Fig 2D), it is likely that changes in SAMD1 chromatin binding have a global influence on the transcriptional landscape in the cells. This hypothesis is supported by the intense dysregulation of cellular pathways during differentiation processes [8] and by the severe phenotype of the SAMD1 knockout mice [16]. It is also notable that the impact of SAMD1 on gene transcription is rather cell type specific. Except for the *L3MBTL3* gene, which appears to be commonly dysregulated upon SAMD1 deletion (Figs 2A and S5G) [8,19], possibly due to a feedback mechanism, most other genes are affected by SAMD1 in a cell type-specific manner [19] (S5G Fig). Thus, the role of SAMD1 is likely highly context dependent, and more work will be necessary to fully understand the specific functions of SAMD1 in the various physiological and pathophysiological contexts.

The study has several limitations. Primarily, most of the work was conducted using human pancreatic ductal adenocarcinoma cell lines, which provide limited insights into the potentially more complex role of SAMD1 in pancreatic cancers in patients. Although our experiments suggest that SAMD1 suppresses EMT-pathway genes in cancer cells, the role of SAMD1 in the development of metastases in patients remains to be explored in further detail. Our work suggests that *CDH2* is the most crucial downstream target of SAMD1 in PDAC cells. We can however not exclude that additional SAMD1 targets are also of relevance for the observed phenotype. Additionally, our work demonstrated that SAMD1 interacts not only with the known partners L3MBTL3, L3MBTL4, and the KDM1A complex, but also with the FBXO11 E3 ubiquitin ligase complex. While our biochemical studies indicate that FBXO11 inhibits the chromatin binding of SAMD1, the precise molecular mechanisms and how FBXO11 influences the biological role of SAMD1 in pancreatic cancer have not been fully elucidated in this study.

Taken together, our work unveiled a tumor-suppressive role of SAMD1 in PDAC through inhibiting EMT-related genes. Exploiting this functionality, particularly by amplifying SAMD1 chromatin binding through the disruption of its interaction with FBXO11 (Fig 6J), holds promise to impede EMT during PDAC progression. Consequently, our findings lay the groundwork for further exploration of SAMD1 as a potential therapeutic target in PDAC and other diseases.

## Material and methods

### Cell culture

Patu8988t cells were cultured in DMEM, high glucose, GlutaMAX Supplement (Thermo Fisher Scientific; 61965026) supplemented with 5% fetal bovine serum (Thermo Fisher Scientific; 10270106). PANC-1 and CFPAC cells were cultured in DMEM, high glucose, GlutaMAX Supplement supplemented with 10% fetal bovine serum, respectively. BxPC3 cells were kept in RPMI 1640 Medium, GlutaMAX Supplement (Thermo Fisher Scientific; 61870036) supplemented with 10% fetal bovine serum, whereas HL-60 cells obtained 15% fetal bovine serum. HEK 293 cells were grown in DMEM/F-12, GlutaMAX Supplement supplemented with 10% fetal bovine serum. HepG2 cells were cultured in MEM, GlutaMAX (Thermo Fisher Scientific; 41090036) supplemented with 10% fetal bovine serum and 1× nonessential amino acids (Thermo Fisher Scientific; 11140050). All cell lines were cultured with 1% penicillin-streptomycin (Thermo Fisher Scientific; 15140122).

### Antibodies

All antibodies used are described in the methods subsections and in S1 Table.

### Stable cell line generation

A SAMD1 knockout was conducted using the Lenti-CRISPR V2 plasmid containing either an unspecific control or guide RNAs targeting *SAMD1* (sg1: AGCGCATCTGCCGGATGGTG; sg2: GAGCATCTCGTACCGCAACG), *CDH2* (sg1: GCCTGAAGCCAACCTTAACTG; sg2: GAGACAATTCAGTAAGCACAG; sg3: GAACTTGCCAGAAAACTCCAG), *FBXO11* (sg1: GAGCCTCTTGTACCCCACCA; sg2: GTGTCCCACAAAGAACAGTA; sg3: GTTTTCTGTAGTTGAAGTTG).

Cells were transfected using Polyethylenimine, Linear, MW 25000, Transfection Grade (Polysciences; 23966), and Opti-MEM (Thermo Fisher Scientific; 31985062). Selection for single clones was performed using 2 μg/μl puromycin (Merck; 58-58-2) for PaTu8988t cells and 0.3 μg/μl for BxPC3 and CFPAC cells. The knockout was confirmed by western blot or immunofluorescence.

To rescue SAMD1, PaTu8988t cells were transfected with SAMD1 constructs containing a FLAG-HA-ER tag. Positive clones were selected using 10 μg/ml blasticidin. After selection, the concentration was reduced to 5 μg/ml blasticidin. Nuclear translocation of FLAG-ER-SAMD1 was induced by adding 200 nM 4-OHT (Merck; 68392-35-8) for 24 h.

### Nuclear extract preparation

To obtain the nuclear extract, the cytoplasmic fraction was removed by incubating harvested cells for 10 min at 4 °C in low salt buffer (10 mM HEPES/KOH (pH = 7.9), 10 mM KCl, 1.5 mM MgCl2, 1xPIC (cOmplete, Protease Inhibitor Cocktail (Roche; 04693116001)), 0.5 mM PMSF). After centrifugation, the remaining pellet was dissolved in high salt buffer (20 mM HEPES/KOH (pH = 7.9), 420 mM NaCl, 1.5 mM MgCl2, 0.2 mM EDTA, 20% glycerol, 1×

PIC, 0.5 mM PMSF) and incubated for 20 min at 4 ˚C while shaking. Subsequently, the lysates were centrifuged, and the supernatant containing the nuclear fraction was further analyzed by western blotting.

## Subcellular fractionation

A subcellular protein fractionation kit for cultured cells (Thermo Fisher Scientific; 78840) was used for fractionation experiments according to the manufacturer's instructions. A 10 cm dish format was applied, which corresponded to a packed cell volume of 20 μl per well.

## Western blot

Western blots were conducted using the Trans-Blot Turbo Transfer System (BioRad; 1704150). The following antibodies were used: anti-tubulin (Merck; MAB3408), anti-SAMD1 antibody (Bethyl; A303-578A), anti-FBXO11 (Novus Biologicals; NB100-59826), anti-KDM1A (Abcam; AB17721), anti-HA (Merck; 11867423), and anti-FLAG (Merck; F3165). Full western blots are presented in S1 Raw Images.

## Immunofluorescence staining

For immunofluorescence staining, cells were seeded on coverslips. On the next day, the cells were fixed with 4% methanol-free formaldehyde (Thermo Fisher Scientific; PI28906) and subsequently permeabilized with 0.5% Triton X-100 in PBS. Blocking was performed with 10% FBS + 0.5% Triton X-100 in PBS. Primary antibody incubation was performed for 1 h in a wet chamber. The following primary antibodies were used at a 1:500 dilution in blocking solution: a homemade SAMD1 antibody recognizing the SAM domain, an HA-antibody (Merck; 11867423), a ZO-1 antibody (Thermo Fisher Scientific; 33–9100), and an N-cadherin antibody (Thermo Fisher Scientific; 33–3900). Next, the cells were washed 3 times with 0.5% Triton X-100 in PBS. Secondary antibody incubation was conducted using Alexa Fluor 488 and 546 coupled antibodies (Thermo Fisher Scientific; A-11008, A-11081; A-11001) at a 1:1,000 dilution. To stain the actin cytoskeleton, cells were stained with 1× Phalloidin-California Red Conjugate (Santa Cruz; sc-499440) for 20 min. Following 3 washing steps, the coverslips were mounted onto microscopy slides using ProLong Diamond mounting medium (Thermo Fisher Scientific; P36961). Photos were taken using a Leica DM 5500 microscope.

## Proliferation assay

To determine proliferation rates, cells were seeded in technical triplicates on 6-well plates at a density of $5 \times 10^4$ cells per well. Cell viability was determined 1, 3, and 7 days after seeding for BxPC3 cells and 1, 3, and 5 days after seeding for PaTu8988t cells using the MTT assay. Therefore, 90 μl of 5 mg/ml thiazolyl blue ≥98% (Carl Roth; 4022) was added to each well. After 1 h, the medium was aspirated, and stained cells were dissolved in 400 μl of lysis buffer (80% isopropanol, 10% 1 M HCl, 10% Triton X-100) and diluted further with PBS if necessary. Absorption was measured at 595 nm using a plate reader. All values were normalized to day 1 to compensate for variations in seeding density. The mean value of 3 biological replicates was determined.

## Wound healing assay

To determine the migration rate of SAMD1 knockout cells, PaTu8988t and BxPC3 cells were seeded in culture inserts (Ibidi; 80209). A total of 70 μl of cell suspension at a density of $6 \times 10^5$ cells per ml was applied. On the next day, the insert was directly removed for BxPC3 cells,

whereas PaTu8988t cells were starved with medium containing 0.5% FBS for 6 h before removing the insert. For rescue experiments, cells were seeded directly in medium containing either ethanol as a solvent control treatment or 200 nM 4-OHT. These treatments were also included during starvation. Photos were taken using an Olympus CKX53 microscope. After 7 h for BxPC3 cells and after 24 h for PaTu8988t cells, photos were taken on the same spots and the cell-free area was measured for both time points using ImageJ Fiji (version: 2.1.0/1.53r).

### Adhesion assay

To investigate the adhesion abilities of the cells, 6-well plates were coated for 2 h at 37 ˚C with 30 µg/ml collagen type I from rat tail (Ibidi; 50201) in 0.02 M acetic acid. Next, the plates were washed 3 times with PBS before $3 \times 105$ PaTu8988t cells per well were applied. Cells attached for 30 min and were subsequently washed 3 times with PBS to remove non-adherent wells before attached cells were counted.

### Invasion assay

To assess the invasion rates of SAMD1 deleted cells, transwell inserts (BD Biosciences; 353097) were coated with 150 µg/ml collagen type I from rat tail (Ibidi; 50204) in the respective medium containing FBS. To allow an even distribution of collagen, 50 µl were added, dispersed, and afterwards 40 µl were removed. The prepared inserts were then positioned in 24-well plates and incubated for 2 h at 37 ˚C. Following this, 100 µl of medium without FBS was introduced into the inserts. PaTu8988t cells were counted in serum-free medium and subsequently stained for 30 min at 37 ˚C with cell tracker (CT) Green (Invitrogen; C2925), 1:1,000 diluted in serum-free medium. After 2 washing steps with serum-free media, $5 \times 104$ cells were applied in a volume of 300 µl to the transwell insert, and 600 µl of medium containing FBS was added to the lower well to allow invasion towards an FBS gradient. Cells invaded for 18 h at 37 ˚C.

On the next day, inserts were washed 3 times with PBS, wiped with a Q-tip and these washing steps were repeated 3 times. To fix the cells, the inserts were inverted, and a few drops of methanol were applied to the filter. After a few minutes, this step was repeated, and bidest water was added. Inserts were wiped out again on the inside and air-dried at room temperature, the filters were then cut out and mounted using Vectashield with DAPI (Vector Laboratories; H-1200) on a microscope slide with a coverslip. Evaluation of migrated tumor cells was performed under a Leica DMI3000B microscope. Migrated cells were counted in 6 visual fields per filter using ImageJ (version: 2.0.0-rc-43/1.52n). Migration was depicted relative to control.

### Transwell migration assay

To determine whether SAMD1 KO has any effects on the migratory potential of PaTu8988t cells, transwell migration assays were performed. Therefore, transwell inserts with a pore size of 8.0 µm (BD Biosciences; 353097) were placed in the wells of a corresponding 24-well plate (Corning; 353504) containing 600 µl serum-free DMEM, high glucose, and GlutaMAX Supplement with or without 5% FBS as a chemoattractant; $2 \times 10^4$ PaTu8988t cells in 300 µl serum-free DMEM medium were seeded per transwell insert. The cells were allowed to migrate through the filter for 18 h. Non-migrated cells were removed from the upper transwell insert by wiping them out and performing thorough washing steps in PBS. The migrated cells present on the bottom side of the transwell filter were fixed in methanol for at least 3 min and stained with crystal violet solution (0.2% in 20% methanol, 1,5 dilution in $dH_2O$) for 10 min at room temperature. Membranes were washed in aqua bidest and dried prior to fixing them on microscopy coverslips using Vectashield with DAPI (Vector Laboratories; H-1200). Evaluation

of migrated tumor cells was performed under a Leica DMI3000B microscope. Migrated cells were counted in 7 visual fields per filter using ImageJ (version: 2.0.0-rc-43/1.52n). Migration was depicted relative to control.

## Time lapse analysis

To perform time lapse analysis, PaTu8988t cells were seeded on collagen-coated 6-well plates at a density of $5 \times 10^4$ cells per well. Coating was performed with 30 mg collagen (Merck; 50201) per ml acetic acid (0.02 M) for 2 h at 37 ˚C. Afterwards, the plates were washed 3 times with PBS before seeding the cells.

On the next day, the cells were placed in a Zeiss LSM 780 microscope and every 10 min photos were taken for 24 h using a 10× DIC objective. Migration of cells was analyzed using the "Time Lapse Analyzer" (University of Ulm, version tla_src_v01_33). As a setup file, "DIC tracking 1" was used. Migration was measured in μm per min.

## Measurement of cell shape

To determine the cell shape, cells were seeded on 6-well plates at low density ($3 \times 10^4$/well). Photos were taken 3 days after seeding. For treatment with ADH-1, cells were seeded on 24-well plates ($1 \times 10^4$) and treated directly after seeding. Photos were taken 1 day later using an Olympus CKX53 microscope. For each condition, 3 photos were taken and 10 cells per photo were analyzed by measuring the circularity of single cells using ImageJ Fiji (version: 2.1.0/1.53r).

## RNA preparation

For RNA isolation, cells were cultivated on 6-well plates up to 80% to 100% confluency. RNA was prepared according to the manufacturer's manual using the RNeasy Mini Kit (Qiagen; 74004) including an on-column DNA digest. For rescue experiments, PaTu8988t cells were seeded at a density of $5 \times 105$ cells per well on a 6-well plate. After 24 h, the medium was exchanged to medium containing either ethanol as a solvent control treatment or 200 nM 4-OHT. Following 24 h, RNA was prepared as described above.

## cDNA synthesis

The Tetro cDNA Synthesis Kit (Bioline; BIO-151 65043) was used to transcribe mRNA into cDNA according to the manufacturer's manual. Samples were incubated at 45 ˚C for 50 min followed by 5 min at 85 ˚C to inactivate Tetro RT. Subsequently, cDNA was diluted 1:20 for use in RT-qPCR.

## RT-qPCR

For analysis by real-time quantitative PCR, MyTaq Mix (Bioline; BIO-25041) was used. For gene expression analysis, values were normalized to GAPDH. Primers are displayed in S2 Table.

## Ectopic co-immunoprecipitation

All ectopic coimmunoprecipitation (Co-IP) experiments were performed in HEK293 cells. Cells were seeded in 10-cm dishes at $2 \times 10^6$ cells per dish. One day later, the expression constructs for 3xHA or FLAG-tagged proteins were transfected using Polyethylenimine, Linear, MW 25000, Transfection Grade (Polysciences; 23966), and Opti-MEM (Thermo Fisher Scientific; 31985062). When the interaction of 2 proteins should be studied in the presence of a

KDM1A inhibitor, the medium was exchanged 5 h after transfection to medium containing either DMSO or 20 nM ORY-1001 (Cay19136; Biomol).

Two days after transfection, extract was prepared using Co-IP buffer (50 mM Tris/Cl (pH = 7.5), 150 mM NaCl, 1% Triton X-100, 1 mM EDTA, 10% glycerol, 1xPIC (cOmplete, Protease Inhibitor Cocktail (Roche; 04693116001)), 0.5 mM PMSF). ANTI-FLAG M2 Affinity Gel (Merck, A2220) beads were equilibrated by washing 2 times with 1× TBS and one time with Co-IP buffer. To bind FLAG-tagged proteins, extracts were added to 50 μl of beads and incubated for approximately 3 h head over tail at 4 ˚C. After incubation, 3 washing steps with Co-IP buffer were performed. The FLAG beads were boiled for 3 min in 2× Laemmli buffer without β-mercaptoethanol. Subsequently, β-mercaptoethanol was added to the supernatant and the samples were analyzed via western blotting.

For Co-IP experiments, the specifics of the used constructs are presented in Tables 1–3, respectively.

**Table 1. SAMD1 constructs.**

| Construct | Amino acids |
| --- | --- |
| FL | 1–538 |
| ΔSAM | 1–450 |
| ΔWH | 111–538 |
| WH | 1–110 |
| SAM | 451–538 |

**Table 2. KDM1A constructs.**

| Construct | Amino acids |
| --- | --- |
| FL | 1–852 |
| IDR/SWIRM/AOD-N | 1–417 |
| Tower | 418–513 |
| AOD-C | 514–852 |
| IDR | 1–172 |
| SWIRM | 173–272 |
| AOD-N | 273–417 |

**Table 3. FBXO11 (Q86XK2-6) constructs.**

| Construct | Amino acids |
| --- | --- |
| FL | 1–843 |
| IDR + F-box | 1–300 |
| repeats | 301–748 |
| ΔUBR | 1–737 |

## Ubiquitination assay

To perform a ubiquitination assay, HEK 293 cells stably overexpressing 3xHA-tagged ubiquitin were seeded on 15 cm dishes and transfected one day afterwards with the respective FLAG-tagged constructs using Polyethylenimine, Linear, MW 25000, Transfection Grade (Polysciences; 23966), and Opti-MEM (Thermo Fisher Scientific; 31985062). Before preparing extracts, cells were treated with 10 µm MG-132 (M7449; Merck) or DMSO as control for 5 h. Extract preparation and IP were performed according to the ectopic co-immunoprecipitation protocol (see above) and samples were analyzed via western blotting.

## Histone demethylase assay

The protocol for a histone demethylase assay was modified after Laurent and colleagues [46]. Wild-type HEK293 cells, HEK293 cells stably overexpressing FLAG-KDM1A and HEK293 cells with SAMD1 KO stably overexpressing FLAG-KDM1A were used. Per reaction, an extract was prepared with buffer A (10 mM HEPES (pH = 7.6), 3 mM $MgCl_2$, 300 mM KCl, 5% glycerol, 0.5% NP-40, 1× PhosSTOP (Roche; 4906845001), 0.5 mM PMSF, 1 µg/µl pepstatin, 10 µg/µl aprotinin, 10 µg/µl leupeptin) using five 15 cm dishes. Subsequently, the extract was diluted by half with buffer B (10 mM HEPES (pH = 7.6), 3 mM $MgCl_2$, 10 mM KCl, 5% Glycerol, 0.5% NP-40, 1× PhosSTOP (Roche; 4906845001), 0.5 mM PMSF, 1 µg/µl Pepstatin I 10 µg/µl Aprotinin I 10 µg/µl Leupeptin), and 50 µl of ANTI-FLAG M2 Affinity Gel (Merck, A2220) beads were equilibrated with buffer B. Extracts were added to the equilibrated beads and incubated for 3 h head over tail at 4 ˚C. Next, the beads were washed 3 times with buffer B and 2 volumes (100 µl) of demethylase buffer (50 mM Tris/Cl (pH = 8.5), 50 mM KCl, 5 mM $MgCl_2$, 0.5% BSA, 5% glycerol, 0.5 mM PMSF, 1 µg/µl pepstatin, 10 µg/µl aprotinin, 10 µg/µl leupeptin, 500 µg/mL FLAG peptide (Merck; F3290)) were added to the beads. To start the demethylase reaction, 3 µg of calf histones (Merck; H9250) was added, and the samples were incubated for 4 h at 37 ˚C while shaking. The reaction was stopped by boiling the samples for 5 min in 5× Laemmli buffer. Subsequently, the samples were analyzed via western blotting.

## Extract preparation and IP for mass spectrometry

For IP mass spectrometry, PaTu8988t cells with SAMD1 KO stably expressing either FH-ER as a control or FH-ER SAMD1 were used. For each construct, twenty 15 cm dishes were seeded and the nuclear translocation of SAMD1 was induced 24 h before extract preparation by adding 200 nM 4-OHT (Merck; 68392-35-8). After collection, cells were centrifuged at 2,000 rpm and 4 ˚C for 10 min. The cell pellet was resuspended in 5× pellet volume hypotonic buffer (10 mM Tris (pH = 7.3), 10 mM KCl, 1.5 mM $MgCl_2$, 0.2 mM PMSF, 10 mM ß-mercaptoethanol, 1xPIC (cOmplete, Protease Inhibitor Cocktail (Roche; 04693116001)), and shaken at 4 ˚C for 10 to 15 min. Next, cells were centrifuged at 2,500 rpm and 4 ˚C for 10 min. The cell pellet was resuspended again in 5× pellet volume hypotonic buffer and denounced 40× in a cell douncer. To remove cell debris, lysates were centrifuged at 3,500 rpm and 4 ˚C for 15 min. The pellet was resuspended in 1× pellet volume low salt buffer (20 mM Tris/Cl (pH = 7.3), 20 mM KCl,

1.5 mM MgCl$_2$, 0.2 mM EDTA, 25% glycerol, 0.2 mM PMSF, 10 mM ß-mercaptoethanol, 1xPIC) and dounced 10×. The sample was shaken in a thermomixer and 0.66× pellet volume of high salt buffer (20 mM Tris/Cl (pH = 7.3), 1.2 M KCl, 1.5 mM MgCl$_2$, 0.2 mM EDTA, 25% glycerol, 0.2 mM PMSF, 10 mM β-mercaptoethanol) was added dropwise. The extract was shaken for 45 min and centrifuged afterwards for 30 min at 13,000 rpm and 4 ˚C.

The supernatant containing the proteins was transferred to a a Slide-A-Lyzer G2 Dialysis Cassette (3.5K) (Thermo Fisher Scientific; 87724) and dialyzed against 3 L of dialysis buffer (20 mM Tris/Cl (pH = 7.3), 100 mM KCl, 0.2 mM EDTA, 20% glycerol, 0.2 mM PMSF, 1 mM DTT) overnight.

To perform the FLAG-IP, the material was retrieved from the dialysis chambers and centrifuged at 13,000 rpm and 4 ˚C for 30 min. Afterwards, a benzonase nuclease (Merck;70664) digest (1 μl of benzonase per 500 μl extract) was performed for 1 h on ice, and 40 μl of ANTI-- FLAG M2 Affinity Gel (Merck, A2220) beads were equilibrated per IP by washing once with TAP buffer (50 mM Tris/Cl (pH = 7.9), 100 mM KCl, 5 mM MgCl$_2$, 0.2 mM EDTA, 10% glycerol, 0.1% NP-40, 0.2 mM PMSF, 1 mM DTT), 3 times with 100 mM glycine (pH = 2.5), once with 1 M Tris/Cl (pH = 7.9) and finally once again with TAP buffer. Subsequently, the extracts were added to the prepared beads and incubated for 3 h, head over tail at 4 ˚C. Afterwards the beads were washed 3× with TAP buffer and 3× with 50 mM ammonium hydrogen carbonate. The washed beads were then sent in for mass spectrometry analysis at the Biomedical Center Munich, protein analysis unit (Head: Axel Imhof), where the enriched proteins were analyzed using a Q Exactive HF Orbitrap Mass Spectrometer, as described previously [10].

### Endogenous Co-IP

For endogenous Co-IP between SAMD1 and KDM1A, an extract was prepared according to the extract preparation protocol for mass spectrometry and the extract was dialyzed overnight (see above). For each IP, one 15 cm dish of PaTu8988t cells was used. Dynabeads Protein A (Thermo Fisher Scientific; 10008D) were equilibrated with TAP buffer (50 mM Tris/Cl (pH = 7.9), 100 mM KCl, 5 mM MgCl$_2$, 0.2 mM EDTA, 10% glycerol, 0.1% NP-40, 0.2 mM PMSF, 1 mM DTT) and subsequently the extract was precleared for 30 min with 10 μl of beads per IP before adding 2 μg of antibody per IP. Self-made IgG and SAMD1 antibodies and an anti-KDM1A (Abcam; AB17721) antibody were applied. After incubation for 3 h, 20 μl of equilibrated Dynabeads Protein A per IP was added and incubated for another 2 h. The beads were washed 3× with TAP buffer before boiling in 2× Laemmli buffer.

### Chromatin preparation

To prepare chromatin, cells were seeded on 15 cm plates at $3 \times 10^6$ cells per plate and cultivated until reaching 70% to 90% confluency. First, 1% formaldehyde was added to the medium and the plates were slowly swayed for 10 min to fix the cells. The fixation was stopped by adding 125 mM glycine for 5 min. Subsequently, the cells were washed twice with PBS and scraped in 1 ml cold buffer B (10 mM HEPES/KOH (pH = 6.5), 10 mM EDTA, 0.5 mM EGTA, 0.25% Triton X-100) per 15 cm plate. All plates containing the same cell line were pooled in a 15 ml tube. The tubes were centrifuged for 5 min at 2,000 rpm and 4 ˚C. The supernatant was removed, and the pellet was resuspended in 1 ml cold buffer C (10 mM HEPES/KOH (pH = 6.5), 10 mM EDTA, 0.5 mM EGTA, 200 mM NaCl) per 15 cm plate followed by a 15-min incubation time on ice. Then, the tubes were centrifuged with the same settings as mentioned before. After removing the supernatant, the pellet was resuspended in 200 μl cold buffer D (50 mM Tris/HCl (pH = 8.0), 10 mM EDTA, 1% SDS, 1xPIC (cOmplete, Protease Inhibitor Cocktail (Roche; 04693116001)) per 15 cm plate, vortexed, and incubated for 10 to

20 min on ice). For shearing the chromatin, the samples were sonicated 2 times for 7 min each using a precooled Bioruptor (Diagenode). The samples were centrifuged for 10 min at 13.000 rpm and 4 ˚C. The supernatant contained the sheared chromatin.

### Chromatin immunoprecipitation

Chromatin immunoprecipitation (ChIP) for ChIP-qPCR was performed according to the One-day ChIP kit protocol (Diagenode; C01010080). Beads were exchanged to Dynabeads Protein A (Thermo Fisher Scientific; 10008D), and Chelex to Chelex 100 Resin (BioRad;142–1253) and ChIP buffer was replaced by a homemade buffer (50 mM Tris/Cl (pH = 7.5), 150 mM NaCl, 5 mM EDT, 1% Triton X-100, 0.5% NP-40). For each ChIP, 3 µg of either IgG control antibody (Diagenode; C15410206) or of a specific antibody were applied. For histone marks only 1 µg of antibody was used. The following antibodies were used: a self-made SAMD1 antibody recognizing the SAM domain, a self-made L3MBTL3 antibody recognizing the SAM domain, anti-KDM1A (Abcam; AB17721), anti-H3K3me2 (Diagenode; C15410035), and anti-H3K4me3 (Diagenode; C15410003).

To prepare samples for ChIP-sequencing, the One-day ChIP kit protocol was used as described above, but the DNA-purification was modified. For DNA elution, beads were incubated with 230 µl elution buffer (100 mM NaHCO$_3$, 1% SDS) for 30 min at room temperature while shaking. Afterward, the samples were centrifuged at 13.000 rpm for 1 min and 200 µl of supernatant was transferred to a fresh tube. The input DNA was dissolved in 50 µl of dH2O and 150 µl of elution buffer was added to obtain an equal volume in all samples, and 8 µl of 5 M NaCl were added to each sample and the samples were incubated at 65 ˚C overnight to reverse the cross-linking.

On the next day, 8 µl of 1 M Tris/Cl (pH = 6.5), 4 µl 0.5 M EDTA, and 2 µl of Proteinase K (10 µg/µl) were added to each sample and all samples were incubated at 45 ˚C for 1 h while shaking. DNA was purified using the QIAquick PCR Purification Kit (Qiagen; 28104) whereby all samples prepared with the same antibody were pooled on the same column. To elute the DNA, columns were incubated for 1 min with 30 µl of sterile 2 mM Tris/Cl (pH = 8.5) and centrifuged at 13.000 rpm for 1 min.

The concentration of the samples was determined using the Quant-iT dsDNA Assay Kit (Thermo Fisher Scientific; Q33120) and the NanoDrop 3300 (Thermo Fisher Scientific). At least 4 ng of DNA was used for library preparation.

Samples were analyzed via ChIP-qPCR using MyTaq Mix (Bioline; BIO-25041). Primers are displayed in S2 Table.

### Library preparation and next-generation sequencing

Next-generation sequencing was performed at the Genomics Core Facility Marburg (Center for Tumor Biology and Immunology, Hans-Meerwein-Str. 3, 35043 Marburg, Germany). For ChIP-seq, the Microplex library preparation kit v2 (Diagenode, C05010012) was used for indexed sequencing library preparation with chromatin immunoprecipitated DNA. Libraries were purified on AMPure magnetic beads (Beckman; A6388). RNA was prepared as described in RNA preparation and integrity was assessed on an Experion StdSens RNA Chip (Bio-Rad; 7007103). RNA-seq libraries were prepared using the TruSeq Stranded mRNA Library Prep kit (Illumina, 2002059). RNA-seq and ChIP-seq libraries were quantified on a Bioanalyzer (Agilent Technologies). Next-generation sequencing was performed on an Illumina NextSeq 550.

## Bioinformatic analyses

ChIP-seq data were mapped to the human genome hg38 using bowtie [47], allowing 1 mismatch. BigWig files were obtained using DeepTools/bamCoverage [48]. Significant peaks were obtained using Galaxy/MACS2 (2.2.7.1) [49]. Heatmaps and profiles were created using Galaxy/DeepTools [48]. The top 250 target genes were identified based on the SAMD1 ChIP-seq signal at promoters. Motif analysis was performed using HOMER [50], comparing SAMD1-bound versus SAMD1-unbound CGIs.

RNA-seq data were aligned to the human transcriptome (GenCode 43) using Galaxy/RNA-Star (2.7.10b) [51]. Differentially expressed genes were obtained using DeSeq2 (2.11.40.7) [52]. Genes with a log2-fold change of more than 0.5 and a $p$-value lower than 0.01 were considered significantly dysregulated. Gene set enrichment analysis was performed using GSEA software with standard settings [23].

The following internet databases and tools were used: Galaxy Europa [53], DeepTools [48], GREAT (4.0.4) [54], Bioconductor/R [55], GSEA (4.3.2) [23], GePIA [21], GDC Data Portal [20], and Kaplan—Meier plotter [22].

The following public ChIP-seq data were used: ChIP-seq of H3K4me3 (GSM945261) [56], H3K27Ac (GSM818826) [57], RNA Polymerase II (GSM1010788) [58] in PANC-1 cells, and ATAC-Seq (GSM1606403) [59] from pancreatic beta cells.

## Statistical analysis

Statistical analysis was performed as described in the figure legends. Error bars indicate standard deviation (SD). The significance of the qPCR results was analyzed via ANOVA or Student's $t$ tests. The significance of changes in SAMD1 ChIP-seq levels was evaluated using a two-sided Kolmogorov—Smirnov test. The significance of the GSEA was evaluated by the GSEA software. All biological experiments were performed in at least 3 replicates. RNA-seq was performed with 3 replicates, using 3 independent SAMD1 KO clones.

## Supporting information

**S1 Fig. SAMD1 has distinct roles in various cancer types.** (A) Expression of *SAMD1* in cancer versus normal tissues. Data from TCGA [20] and visualized via GePIA [21]. Cancer types highlighted in red indicate significantly up-regulated *SAMD1* expression. (B) Kaplan—Meier survival curves (overall survival) in liver hepatocellular carcinoma (LIHC) and kidney renal clear cell carcinoma (KIRC), using auto-selected cut-offs. High SAMD1 expression correlates with a worse prognosis. (C) Kaplan—Meier survival curves (overall survival) in cervical cancer (CESC) and thymoma (THYM). High *SAMD1* expression correlates with a better prognosis. Data in (B) and (C) are derived from TCGA and visualized via the Kaplan—Meier plotter tool [22] using auto-selected cut-off. (D) GSEA of the epithelial—Mesenchymal transition (EMT) pathway in the cancer types presented in (B) and (C). (E) GSEA of the MYC target genes in the cancer types presented in (B) and (C) and of PDAC. In (D) and (E), tissue samples with high *SAMD1* expression are compared to samples with low *SAMD1* expression. The data underlying this figure is available in S1 Data.
(TIF)

**S2 Fig. Transwell, adhesion, invasion, and time-lapse assays in PaTu8988t cells.** (A) Transwell migration assay of PaTu8988t control and SAMD1 KO cells. Data represent the mean ± SD of 3 biological replicates. Significance was analyzed using Student's $t$ test. (B) Representative crystal violet staining of 1 transwell migration assay. (C) Migration of PaTu8988t control and SAMD1 KO cells in μm/min based on time-lapse analysis. See also S1 and S2

Movies. Data represent the mean ± SD of 3 biological replicates. Significance was analyzed using Student's *t* test. (D) Adhesion assay of PaTu8988t control and SAMD1 KO cells. Data represent the mean ± SD of 3 biological replicates. Significance was analyzed using Student's *t* test. (E) Invasion assay of PaTu8988t control and SAMD1 KO cells. Data represent the mean ± SD of 2 biological replicates. Significance was analyzed using Student's *t* test. (F) Representative cell tracker staining of 1 adhesion assay. The data underlying this figure is available in S1 Data.
(TIF)

**S3 Fig. Effect of SAMD1 KO on BxPC3 cells.** (A) Western blot showing BxPC3 wild-type cells, control cells, and 2 different SAMD1 knockout clones. (B) Proliferation assay of BxPC3 wild-type cells, control cells, and 2 different SAMD1 knockout clones. Data represent the mean ± SD of 3 biological replicates. Significance was analyzed using one-way ANOVA. (C) Representative picture of one wound healing assay of BxPC3 control cells and one SAMD1 knockout clone. (D) Quantification of the wound healing assay from (C). Data represent the mean ± SD of 3 biological replicates, and significance was analyzed using Student's *t* test. The data underlying this figure is available in S1 Data.
(TIF)

**S4 Fig. Validation of the ER-SAMD1 fusion protein.** (A) Immunofluorescence of PaTu8988t SAMD1 knockout cells with or without induction of SAMD1 rescue, Bar = 20 μm. (B) Western blot after fractionation of PaTu8988t SAMD1 knockout cells with or without induction of SAMD1 rescue. (C) Quantification of (B). (D) Representative picture of one wound healing assay of PaTu8988t SAMD1 KO cells expressing FH-ER with and without 4-OHT induction. (E) Quantification of the wound healing assay from (D). Data represent the mean ± SD of 2 biological replicates, and significance was analyzed using Student's *t* test. The data underlying this figure is available in S1 Data.
(TIF)

**S5 Fig. Transcriptional regulation by SAMD1 in PaTu8998t cells.** (A) Principal component analysis (PCA) of RNA-Seq data upon SAMD1 KO. Three clonally independent SAMD1 KO clones were used. (B) Heatmap of the significantly dysregulated genes. Examples of the most dysregulated genes are shown on the right. (C) GSEA analysis of mesenchymal-related pathways in SAMD1 KO versus control cells (D). Heatmap of genes in genesets from GSEA analysis in Figs 2C and S5C. The core enriched genes are marked. (E) Comparison of gene expression of EMT-related genes in control and SAMD1 KO cells, based on RNA-Seq data. *P*-values are deried from DeSeq2. (F) RT-qPCR of EMT-related genes in BxPC3 or CFPAC cell lines comparing SAMD1 KO versus control cells. Data represent the mean ± SD of 3 biological replicates. Significance was analyzed using Student's *t* test. (G) Expression of *TJP1* in control and SAMD1 KO PaTu8998t cells based on RNA-Seq (upper panel) and immunofluorescence (lower panel). (H) Comparison of gene expression changes in PaTu8988t cells versus mouse ES cells [8] and HepG2 cells [19]. *L3MBTL3* is the only gene that is consistely up-regulated. (I) Comparison of GSEA results from mesenchymal-related pathways in PaTu8988t, mouse ES [8], and HepG2 [19] cells upon SAMD1 KO. The data underlying this figure is available in S1 Data.
(TIF)

**S6 Fig. Detailed analysis of SAMD1 chromatin binding.** (A) Heatmaps of SAMD1-bound and -unbound CGIs regarding RNA Polymerase II, H3K4me3, H3K27ac, and ATAC-Seq. The heatmaps base on public ChIP-seq data, which can be found in the GEO database with accession numbers GSM945261, GSM818826, GSM1010788, and GSM1606403. (B) Gene ontology

analysis of SAMD1 genomic targets using GREAT [54]. (C) Comparison of SAMD1 promoter level versus gene expression changes upon SAMD1 KO. Significant up- and down-regulated genes (cut-off: log2-fold-change > 0.5; *p*-value < 0.01) are colors in red and blue, respectively. (D) Promoter profiles of SAMD1 at the significant dysregulated genes from (C). The data underlying this figure is available in S1 Data.
(TIF)

**S7 Fig. CDH2 KO or inhibition rescues the migration phenotype upon SAMD1 deletion.** (A) Cell shape of control, CDH2 KO, SAMD1 KO, and CDH2/SAMD1 double KO PaTu8988t cells. Circularity was determined using ImageJ Fiji. Significance was analyzed using one-way ANOVA. (B) Example bright field microscopy for (A). (C) Cell shape of PaTu8988t wild-type cells, control cells, and 2 different SAMD1 knockout clones with or without application of the N-cadherin inhibitor ADH-1. Circularity was determined using ImageJ Fiji. Significance was analyzed using one-way ANOVA. (D) Example bright field microscopy for (C). The data underlying this figure is available in S1 Data.
(TIF)

**S8 Fig. Effects of SAMD1 at the L3MBTL3 promoter and in BxPC3 cells.** (A) RT-qPCR measuring *L3MBTL3* expression with or without induction of SAMD1 rescue in PaTu8988t control and SAMD1 KO cells. WHmut = R45A/K46A mutation of SAMD1 [8]. Data represent the mean ± SD of 4 biological replicates. Significance was analyzed using Student's *t* test. (B) SAMD1 ChIP-qPCR at the *L3MBTL3* promoter with or without induction of SAMD1 rescue in PaTu8988t control and SAMD1 KO cells. Data represent the mean ± SD of 3 biological replicates. Significance was analyzed using Student's *t* test. (C) ChIP-qPCR at the *CDH2* promoter in BxPC3 control and SAMD1 KO cells using IgG or SAMD1 antibodies. Data represent the mean ± SD of 2 biological replicates. Significance was analyzed using Student's *t* test. RT-qPCR measuring *CDH2* expression in BxPC3 wild-type cells, control cells, and 2 different SAMD1 knockout clones. Data represent the mean ± SD of 3 biological replicates. Significance was analyzed using one-way ANOVA. (D) ChIP-qPCR at the *L3MBTL3* promoter in BxPC3 control cells and SAMD1 KO cells, using SAMD1 or IgG antibodies. Data represent the mean ± SD of 2 biological replicates. Significance was analyzed using Student's *t* test. RT-qPCR measuring *L3MBTL3* expression in BxPC3 wild-type cells, control cells, and 2 different SAMD1 knockout clones. Data represent the mean ± SD of 3 biological replicates. Significance was analyzed using one-way ANOVA. The data underlying this figure is available in S1 Data.
(TIF)

**S9 Fig. SAMD1/KDM1A interaction is influenced by SAM domain and ORY-1001.** (A) Co-immunoprecipitation in HEK293 cells showing the interaction between SAMD1 full-length or SAMD1 ΔSAM and the KDM1A-complex. (B) Co-immunoprecipitation in HEK293 cells showing the interaction between SAMD1ΔSAM and KDM1A upon treatment with the KDM1A inhibitor ORY-1001. (C) Co-immunoprecipitation in HEK293 cells showing the interaction between PHF21A and KDM1A upon treatment with the KDM1A inhibitor ORY-1001. (D) Co-immunoprecipitation in HEK293 cells showing the interaction between RCOR1 and KDM1A upon treatment with the KDM1A inhibitor ORY-1001. (E) Co-immunoprecipitation in HEK293 cells showing the interaction between FBXO11 and KDM1A upon treatment with the KDM1A inhibitor ORY-1001. (F) Co-immunoprecipitation in HEK293 cells showing the interaction between full-length FBXO11 and FBXO11 ΔUBR with SAMD1. (G) Co-immunoprecipitation in HEK293 cells showing the interaction between full-length FBXO11 and FBXO11 ΔUBR with KDM1A.
(TIF)

**S10 Fig. Detailed analysis of the consequence of FBXO11 deletion on SAMD1 chromatin binding.** (A) Violin plots showing the SAMD1 level in PaTu8988t control and FBXO11 KO cells at SAMD1 bound locations for both ChIP-Seq replicates. Statistical significance was evaluated using a Kolmogorov—Smirnov test. (B) SAMD1 levels at the 4 different groups identified in Fig 6C in PaTu8988t control and FBXO11 KO cells (from replicate 1). (C) CpG island size of 4 different groups identified in Fig 6C. Statistical significance was evaluated using a Kolmogorov—Smirnov test. (D) Heatmap of H3K4me3 and Pol II in the 4 different groups identified in Fig 6C. The heatmaps base on public ChIP-seq data, which can be found in the GEO database with accession numbers GSM945261 and GSM1010788. (E) Profiles of H3K4me3 and RNA Polymerase II at the 4 different groups identified in Fig 6C. (F) Expression of genes in the 4 different groups identified in Fig 6C. (G) Representative picture of a wound healing assay of PaTu8988t wild-type cells, control cells, and 2 different FBXO11 knockout clones. (H) Quantification of (G). Data represent the mean ± SD of 2 biological replicates. Significance evaluated using one-way ANOVA. The data underlying this figure is available in S1 Data. (TIF)

**S11 Fig. SAMD1 chromatin binding upon FBXO11 overexpression in HEK293 cells.** (A) ChIP-qPCR of FBXO11 and SAMD1 after FBXO11 overexpression in HEK293 cells. Data represent the mean ± SD of 3 biological replicates. Significance was analyzed using Student's *t* test. (B) Relative expression of SAMD1 target genes from (A) upon FBXO11 overexpression measured by RT-qPCR. Data represent the mean ± SD of 3 biological replicates. The data underlying this figure is available in S1 Data. (TIF)

**S12 Fig. Comparison of the role of SAMD1 and FBXO11 in PDAC, Thymoma and CESC using data from TCGA.** (A) GSEA of the epithelial—Mesenchymal transition (EMT) pathway comparing patient samples with SAMD1 high versus SAMD1 low and FBXO11 high versus FBXO11 low. (B) Kaplan—Meier survival curves (overall survival) based on SAMD1 and FBXO11 expression. Data are derived from TCGA and visualized via the Kaplan—Meier plotter tool [22] using auto-selected cut-off. Some of the results are already shown in Figs 1, 6 and S1, but are included here for comparison. The data underlying this figure is available in S1 Data. (TIF)

**S1 Table. Used antibodies.** (XLSX)

**S2 Table. Used RT-PCR primers.** (XLSX)

**S1 Movie. The 24-h time lapse video of migration of PaTu8988t control cells.** Photos were taken every 10 min. (MP4)

**S2 Movie. The 24-h time lapse video of migration of PaTu8988t SAMD1 KO cells.** Photos were taken every 10 min. (MP4)

**S1 Data. Excel spreadsheet containing, in separate sheets, the underlying numerical data for Figs 1A, 1E, 1G, 1I, 1K, 2G, 2K, 2L, 2M, 3A, 3B, 3H, 4B, 5D, 6A, 6B, 6E and 6G as well as S1A, S1B, S1C, S2A, S2C, S2D, S2E, S3B, S3D, S4C, S4E, S5A, S5B, S5D, S5E, S5F, S5G, S5H, S5I, S6B, S6C, S7A, S7C, S8A, S8B, S8C, S8D, S10A, S10B, S10C, S10E, S10H, S11A,**

**S11B** and **S12B** Figs.
(XLSX)

**S1 Raw Images. This file contains all raw images of western blots.**
(PDF)

## Acknowledgments

We acknowledge the Protein Analytics Unit at the Biomedical Center, Ludwig-Maximilians University Munich, for providing services and assistance with data analysis. We thank Matthias Lauth and Uta-Maria Bauer for providing PDAC cell lines and for discussions.

## Author Contributions

**Conceptualization:** Clara Simon, Robert Liefke.

**Data curation:** Inka D. Brunke, Bastian Stielow, Ignasi Forné, Anna Mary Steitz, Merle Geller, Iris Rohner, Lisa Marie Weber, Sabrina Fischer, Lea Marie Jeude, Theresa Huber, Andrea Nist, Robert Liefke.

**Formal analysis:** Inka D. Brunke, Ignasi Forné, Merle Geller, Sabrina Fischer, Robert Liefke.

**Investigation:** Clara Simon, Robert Liefke.

**Methodology:** Clara Simon.

**Project administration:** Robert Liefke.

**Supervision:** Thorsten Stiewe, Magdalena Huber, Malte Buchholz, Robert Liefke.

**Validation:** Robert Liefke.

**Visualization:** Clara Simon, Robert Liefke.

**Writing – original draft:** Clara Simon, Robert Liefke.

**Writing – review & editing:** Clara Simon, Robert Liefke.

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
