## [Editor Report · Decision Letter 0]

2 Nov 2023

Dear Dr Liefke, 

Thank you for submitting your manuscript entitled "SAMD1 suppresses epithelial-mesenchymal transition (EMT) pathways in pancreatic ductal adenocarcinoma" for consideration as a Research Article by PLOS Biology. Please accept my apologies for the delay in getting back to you as we consulted with an academic editor about your submission. 

Your manuscript has now been evaluated by the PLOS Biology editorial staff, as well as by an academic editor with relevant expertise, and I am writing to let you know that we would like to send your submission out for external peer review.

Once your full submission is complete, your paper will undergo a series of checks in preparation for peer review. After your manuscript has passed the checks it will be sent out for review. To provide the metadata for your submission, please Login to Editorial Manager (https://www.editorialmanager.com/pbiology) within two working days, i.e. by Nov 04 2023 11:59PM.

Kind regards,

Richard

Richard Hodge, PhD

rhodge@plos.org

PLOS

---

## [Decision Letter · Decision Letter 1]

22 Dec 2023

Dear Dr Liefke,

Thank you for your patience while your manuscript "SAMD1 suppresses epithelial-mesenchymal transition (EMT) pathways in pancreatic ductal adenocarcinoma" was peer-reviewed at PLOS Biology. Please accept my apologies for the delays that you have experienced during the peer review process. Your manuscript has been evaluated by the PLOS Biology editors, an Academic Editor with relevant expertise, and by three independent reviewers.

As you will see in the reviewer reports, which can be found at the end of this email, the reviewers are generally positive about your manuscript and think it is interesting and well conducted. However, Reviewers #1 and #3 both raise overlapping concerns with the overall strength of the mechanistic insights regarding the SMAD1/FBOX11 interaction and its role in EMT and tumor aggressiveness. Reviewer #3 also notes that a more comprehensive assessment of the effect of SMAD1 on EMT should be added in the revised version, including additional functional assays and confirming a larger panel of EMT genes. 

Based on their specific comments and following discussion with the Academic Editor, it is clear that a substantial amount of work would be required to meet the criteria for publication in PLOS Biology. However, given our and the reviewer interest in your study, we would be open to inviting a comprehensive revision of the study that thoroughly addresses all the reviewers' comments. Given the extent of revision that would be needed, we cannot make a decision about publication until we have seen the revised manuscript and your response to the reviewers' comments. Your revised manuscript would need to be seen by the reviewers again, but please note that we would not engage them unless their main concerns have been addressed. 

We appreciate that these requests represent a great deal of extra work, and we are willing to relax our standard revision time to allow you 6 months to revise your study. Please email us (plosbiology@plos.org) if you have any questions or concerns, or envision needing a (short) extension.

**IMPORTANT - SUBMITTING YOUR REVISION**

*Resubmission Checklist*

*Published Peer Review*

*PLOS Data Policy*

*Blot and Gel Data Policy*

Sincerely,

Richard

Richard Hodge, PhD

rhodge@plos.org

REVIEWS:

Reviewer #1: In the manuscript by Simon et al. the authors addressed the role of the CpG island binding protein SAMD1 in pancreatic cancer. They find that it is upregulated in several cancers, but displays potentially opposing roles in cancer as its expression either correlates or anticorrelates with prognosis. In PDAC SAMD1 was found to repress genes associated with epithelial-mesenchymal transition (like N-cadherin) and knockout of SAMD1 in PaTu8988t and BxPC3 cell lines resulted in derepression to unleash migration. The authors found that SAMD1 function depends on its interaction with KDM1A and FBXO11, suggested to form a ternary complex that is recruited to CpG islands at target promoters.

We lack of sufficient mechanistic insight how deregulated gene expression rapidly renders malignant transformation and therapy resistance. Hence, identifying exploitable vulnerabilities is of high relevance. Therefore, the manuscript is of potential interest and merits to be published in PLoS Biology. The analyses are strong in dissecting the mechanism how SAMD1 is interacting with cofactors and how this impacts regulation of target genes. In particular, which protein domains of SAMD1, FBXO11 and KMD1A are involved in binding was done very accurately. Where the story falls short is in understanding how SAMD1 can have context-dependent protumorigenic and antitumorigenic functions. Also the analysis of the role of the SAMD1-FBXO11 interaction is a bit descriptive only. This should be improved in a revised version. Until now, the FBXO11 part is a bit isolated and the consequence of FBXO11 loss remains unknown. I listed my specific comments below:

1. Repression of CDH2 is a main finding to enable EMT and cell motitity. However, mechanistic dissection revealed that in the end KO of FBXO11, as novel interaction partner of SAMD1, was not so much affecting CDH2 expression (and of the bona fide target L3MBTL3), although FBXO11 KO results in global increase of SAMD1 chromatin binding and in particular at the CDH2 promoter. So there is something puzzling that does not really fit. Based on the model in Fig. 6i,j the complexes are mutually exclusive, which the data do not support (in fact more the opposite?!). Even if there are different complexes, the FBXO11 function does not affect any of the previously identified targets. The only link between SAMD1 and FBXO11 function is that the EMT score of FBXO11-hi vs. low tumor samples from TCGA in Fig. 6h is increased in FBXO11-hi tumors, which might be completely independent from SAMD1.

2. In line with that, the statement in lines 495-499 "…types, such as diffuse large B-cell lymphomas and lung cancer [31,34]. In lung cancer cell lines, the FBXO11-containing complex was found to neddylate p53, thereby inhibiting its transcriptional activity." suggests that the functional connection of SAMD1 and FBXO11 cannot be clearly separated from other independent functions of FBXO11. Likely, there are overlapping effects and apparently the SAMD1 specific effect of FBXO11 is minor. Moreover, since the PaTu8988t cell line is mutant for p53, loss or gain of function of mutant neddylated p53 or effects of FBXO11/p53 binding could mainly influence the cellular phenotype, independent of SAMD1.

3. Fig. 3b,4f: There is no detectable expression of Tower and AOD_C in input detected and the IP is also not convincing for these fragments, so it is difficult to draw robust conclusions from this experiment. Especially in the 'Tower sample: Which of the many bands in the IP is actually the IP'd protein? This needs to be improved. Similarly, in lines 337-339: "This interaction is facilitated by the N-terminal part of the AOD domain of KDM1A, which is the same region as for the interaction with SAMD1 (Figure 4f, 3f)." This conclusion is not very convincing, since AOD_C and AOD_N are hardly expressed or IP'd. So the findings should be confirmed by a SAMD1delAOC-N vs. SAMD1delAOC-C construct. An additional point: I can see the non-cropped immunoblots in the supplement with MW indicators, but in particular for the IPs with the fragments of different size, MW indicators would be helpful.

4. Lines 283-285, Fig. 3g,h: "This result suggests that SAMD1 modulates the function of KDM1A, possibly not just by influencing its recruitment to chromatin but also by influencing the catalytic activity of KDM1A." I wonder how this assay can discriminate whether the reduced me2 H3K4 levels are due to reduced histone/chromatin binding or reduced enzymatic activity? 

Minor points:

5. Some of the analysis are not significant, this should be indicated in the text. Crucially, EMT signatures GSEA is not significant upon SAMD1 KO (Fig. 2c). Here maybe it helps to discriminate between up and down regulated genes, as e.g. Cdh1/Cdh2 would be counter regulated.

6. Are Fig. 2d,g and S4c the only result that can be extracted from the SAMD1 ChIP-Seq? It would be worth to correlate differential expression with SAMD1 peaks at CpG islands of the corresponding promoters.

7. lines 258-261: The authors claim that "Consistently, via ChIP-qPCR, we observed an increase in H3K4me2 and H3K4me3 at the L3MBTL3 and CDH2 gene promoters upon SAMD1 deletion (Figure 3b)." However, the data indicate only an increase in me3 at the CHD2 promoter.

8. How are the 250 SAMD1 target genes defined?

9. The supplemenetal figures need to be reorganized in the correct order: (2a-g, 3a-b), 4, 5, (3c-d, 2h-i)

Reviewer #2: I would like to congratulate the authors for the intense job. Here you will finde my comments hoping they improve your manuscript.

Minor revisions

About figure 1a. Around 20 months the difference among SMAD1 expression levels becomes relevant, but not before, regarding survival (if probability in Y exes means probability to be alive). Could the authors discuss this observation considering SMAD1 as a tumor progression marker?

Major revisions

The quality of the figures is bad. I can't properly see the cells and descriptions. The quality of supplemental material is good.

About Figure 1J. I am concerned about the rescue experiment using ER-SAMD1 fusion protein. How would you discard that migration properties do not belong to the ER part of the fusion protein?

How could the authors evaluate that the proteins determined in Figure 4 (b,c) are not bound to the ER part of the protein?

Reviewer #3: In the manuscript by Liefke and colleagues, the authors examine the role of SAMD1 as a suppressor of tumor aggressiveness in PDAC at least in part via suppressing EMT as measured primarily through examining the levels of two EMT associated genes and examining migration. They also go on to show that SAMD1 is negatively regulated by FBXO11. The function of SAMD1 in PDAC appears to be different then that in other tumor contexts (where it is tumor promotional) and this finding, coupled with the FBXO11 regulation of SAMD1 provides novelty to the work. Overall, the experiments are rigorously performed and the data are well interpreted. Addressing the following issues would help to strengthen the paper:

1. Since the major claim is that SAMD1 regulates EMT, a more comprehensive analysis of EMT is required. The authors should confirm a larger panel of EMT genes than just CDH2 and should also perform other studies of functionality associated with EMT (invasion, adhesion, polarity etc). The authors should also present the heat map of transcriptional changes in the EMT pathway with SAMD1 KO. Making a broad emt claim without more careful analysis is not warranted.

2. The ER-fusion for SAMD1 does not appear to affect nuclear localization as much as binding to chromatin (based on the fractionation in suppl fig3B). Mechanistically- how is this working? 

3. Is it possible to show SAMD1 knock out is necessary for the EMT phenotypes by trying to induce an EMT with specific stimuli in the context of control vs KO cells? Can SAMD1 function in PDAC cells to prevent multiple forms of EMT induction?

4. Why does increased binding to chromatin with FBXO11 deletion only impact specific genes (a small subset)? How does this relate to the EMT phenotypes (are the genes transcriptionally changed mostly EMT genes)? Is there any identifying factor (sequence specificity- poteinally other binding partners) that would explain why only a small subset of genes in which binding changes are transcriptionally changed? 

5. How does FBXO11 affect EMT phenotypes (Migration etc)? This can be tested with the FBXO11 depleted cells. 

Overall, there is significant novelty in working out this pathway but the paper remains incomplete in its assessment of how the SAMD1/FBXO11 axis is impinging on EMT to influence tumor aggressiveness.

---

## [Decision Letter · Decision Letter 2]

16 May 2024

Dear Dr Liefke,

Thank you for your patience while we considered your revised manuscript "SAMD1 suppresses epithelial-mesenchymal transition (EMT) pathways in pancreatic ductal adenocarcinoma" for consideration as a Research Article at PLOS Biology. Please accept my apologies for the delays that you have experienced during this round of the peer review process. Your revised study has now been evaluated by the PLOS Biology editors, the Academic Editor and the original reviewers. 

In light of the reviews, which you will find at the end of this email, we are pleased to offer you the opportunity to address the remaining points from Reviewer #3 in a revision that we anticipate should not take you very long. This includes providing additional data (such as qPCR) in other pancreatic cell lines to bolster the claims that SAMD1 has an anti-EMT function. We also strongly encourage you to provide additional evidence for the functional consequences for the regulation of SAMD1 chromatin binding by FBX011 (comment 4). We will then assess your revised manuscript and your response to the reviewers' comments with our Academic Editor aiming to avoid further rounds of peer-review, although might need to consult with the reviewers, depending on the nature of the revisions.

We expect to receive your revised manuscript within 2 months. Please email us (plosbiology@plos.org) if you have any questions or concerns, or would like to request an extension. 

**IMPORTANT - SUBMITTING YOUR REVISION**

*Resubmission Checklist*

*Published Peer Review*

*PLOS Data Policy*

*Blot and Gel Data Policy*

Sincerely,

Richard

Richard Hodge, PhD

rhodge@plos.org

REVIEWS:

Reviewer #1: The authors addressed all my comments and improved the manuscript significantly. I like to congratulate the authors for their very nice work.

Reviewer #2: Dear authors.

Thanks for considering all my request. Changes are sufficient for publication regarding my concerns. Good luck with the rest of the evaluation.

Best,

Reviewer #3: Overall, the authors have addressed most of my concerns from the first version of this manuscript. A few minor concerns remain:

1. The authors are arguing for context specific functions of SAMD1 in pancreatic cancer with respect to EMT, but only show effects on EMT gene signatures in one pancreatic cancer line (but not in other non-pancreatic cancer lines tested). To argue that SAMD1 has a specific anti-EMT function in pancreatic cancer, the authors should (at a minimum), show that they can identify some of the same EMT gene expression changes in other pancreatic cancer cell lines by qRT-PCR.

2. Line 465 addresses enrichment of NEDD8, but I do not see this data in the figure.

3. Line 554 says that increased binding of SAMD1 at the CDH2 promoter in the FBX011 KO cells- but this is not a significant change and thus the wording needs to be tempered.

4. While the authors do provide convincing data that FBX011 affects SAMD1 binding to chromatin, there is no clear functional consequence of this (at the gene expression or phenotypic level). This remains unfortunately one of the weaker parts of the manuscript- as it is unclear whether the effects of FBX011 on SAMD1 have any real consequences.

5. Line 651, I believe the authors meant "high FBX011" not "SAMD1"

---

## [Editor Report · Decision Letter 3]

21 Jun 2024

Dear Dr Liefke,

Thank you for your patience while we considered your revised manuscript "SAMD1 suppresses epithelial-mesenchymal transition (EMT) pathways in pancreatic ductal adenocarcinoma" for publication as a Research Article at PLOS Biology. This revised version of your manuscript has been evaluated by the PLOS Biology editors and the Academic Editor.

Based on our Academic Editor's assessment of your revision", we are likely to accept this manuscript for publication, provided you satisfactorily address the following data and other policy-related requests.

IMPORTANT-please attend to the following:

A) We would like to suggest a different title to improve accessibility. Please change your title to: “SAMD1 suppresses epithelial-mesenchymal transition pathways in pancreatic ductal adenocarcinoma”.

B) Please make the deposited data publicly available. We have noticed that you have deposited your data, but that they currently are private. These need to be publicly available before the manuscript enters production. 

C) DATA POLICY:

Regardless of the method selected, please ensure that you provide the individual numerical values that underlie the summary data displayed in the following figure panels as they are essential for readers to assess your analysis and to reproduce it: Fig. 1A, Fig. S1A-C, Fig. S2E, Fig. S5C, Fig. S6A,D, Fig. S10A-C, EF and Fig. S12B

Please also ensure that figure legends in your manuscript include information on where the underlying data can be found, and ensure your supplemental data file/s has a legend. This includes references to repositories or papers for publicly available data. 

D) CODE POLICY

Please note that we cannot accept sole deposition of code in GitHub, as this could be changed after publication. However, you can archive this version of your publicly available GitHub code to Zenodo. Once you do this, it will generate a DOI number, which you will need to provide in the Data Accessibility Statement (you are welcome to also provide the GitHub access information). See the process for doing this here: https://docs.github.com/en/repositories/archiving-a-github-repository/referencing-and-citing-content.

We expect to receive your revised manuscript within two weeks. 

*Published Peer Review History*

*Press*

Sincerely,

Suzanne

Suzanne De Bruijn, PhD, 

Associate Editor

sbruijn@plos.org

PLOS Biology

---

## [Editor Report · Decision Letter 4]

5 Jul 2024

Dear Dr Liefke,

Thank you for the submission of your revised Research Article "SAMD1 suppresses epithelial-mesenchymal transition pathways in pancreatic ductal adenocarcinoma" for publication in PLOS Biology. On behalf of my colleagues and the Academic Editor, Albana Gattelli, I am pleased to say that we can in principle accept your manuscript for publication, provided you address any remaining formatting and reporting issues. These will be detailed in an email you should receive within 2-3 business days from our colleagues in the journal operations team; no action is required from you until then. Please note that we will not be able to formally accept your manuscript and schedule it for publication until you have completed any requested changes.

PRESS

Sincerely, 

Suzanne

Suzanne De Bruijn, PhD, 

Associate Editor

PLOS Biology

sbruijn@plos.org